# Toward Practical Equilibrium Propagation: Brain-inspired Recurrent Neural Network with Feedback Regulation and Residual Connections

**Zhuo Liu**
School of Microelectronics
University of Science and Technology of China
Hefei 230026, Anhui, China
zhuoliu00@mail.ustc.edu.cn

**Tao Chen** *
School of Microelectronics
University of Science and Technology of China
Hefei 230026, Anhui, China
tchen@ustc.edu.cn

## Abstract

Brain-like intelligent systems need brain-like learning methods. Equilibrium Propagation (EP) is a biologically plausible learning framework with strong potential for brain-inspired computing hardware. However, existing implementations of EP suffer from instability and prohibitively high computational costs. Inspired by the structure and dynamics of the brain, we propose a biologically plausible Feedback-regulated REsidual recurrent neural network (FRE-RNN) and study its learning performance in the EP framework. Feedback regulation enables rapid convergence by attenuating feedback signals and reducing the disturbance of feedback paths to feedforward paths. The improvement in the convergence property reduces the computational cost and training time of EP by orders of magnitude, delivering performance on par with backpropagation (BP) in benchmark tasks. Meanwhile, residual connections with brain-inspired topologies help alleviate the vanishing gradient problem that arises when feedback pathways are weak in deep RNNs. Our approach substantially enhances the applicability and practicality of EP. The techniques developed here also offer guidance for implementing in-situ learning in physical neural networks.

## 1 Introduction

Backpropagation (BP) has been the driving force behind the success of artificial intelligence (AI) across a wide variety of tasks, ranging from image recognition to natural language processing (Rumelhart et al., 1986; Lecun, 1988; He et al., 2016; Vaswani et al., 2017). Despite these triumphs, BP's reliance on non-local error signals and weight transport lacks biological plausibility (Journé et al., 2023; Ororbia, 2023). The brain does not appear to implement the gradient computations performed by BP, in particular the explicit derivative of the activation function, which demands precise access to the rate of change in neuronal activities at specific operating points (Ororbia, 2023). Moreover, implementing BP in neuromorphic systems incurs enormous overhead (Kudithipudi et al., 2025). Drawing inspiration from the topology and dynamics of the brain is a viable approach to advancing biologically plausible learning mechanisms and to promoting energy-efficient computing systems for AI.

Equilibrium Propagation (EP) (Scellier & Bengio, 2017; Ernoult et al., 2019; Laborieux et al., 2021) presents a compelling and hardware-friendly alternative. It leverages naturally settling dynamics in RNN for credit assignment, and eliminates the need for explicit activation derivatives. EP operates in two phases with nearly identical dynamics, and the synaptic adjustments depend only on local information (Ackley et al., 1985; Movellan, 1991; Ernoult et al., 2020). In EP, the output layer is softly nudged by the prediction error toward configurations that incrementally minimize the loss function, a regime termed weak supervision (Millidge et al., 2023). A major drawback of EP is

---

*Corresponding Author

its notably slow training speed and instability. An RNN often requires dozens or even hundreds of iterations to reach a stable state (Scellier & Bengio, 2017). Previous attempts to optimize EP's performance have led to markedly more complicated procedures (O'Connor et al., 2019; Laborieux & Zenke, 2024).

In this paper, we draw inspiration from the brain and propose a Feedback-regulated REsidual recurrent neural network (FRE-RNN). We substantially improve the convergence properties of the RNNs and training speed of EP while achieving performance comparable to BP. Our contributions are as follows:

- By scaling down the feedback strength of RNNs, we enhance the robustness of EP and accelerate the training and inference speed by orders of magnitude because of the improved convergence properties.

- To counteract the gradient vanishing problem caused by weak feedback, we introduce residual connections into the layered RNNs, enabling the training of deep networks that previously challenged EP and achieving performance closer to BP.

- The feedback regulation and residual connections in RNNs of arbitrary graph topologies mirror the multi-scale recurrence in biological neural networks. Our work fosters EP's biological plausibility and extends its applicability in brain-inspired computational hardware.

## 2 BACKGROUND

### 2.1 CONVERGENT RNNS WITH STATIC INPUT

Consider an RNN as a dynamical system driven by a static input $x$:

$$s[t+1] = F(x, s[t], \theta), \tag{1}$$

where $F$ is the transition function, $s[t]$ is the network state at time step $t(t = 0, 1, 2, \ldots, T)$, and $\theta$ denotes the parameters. Assuming that the network state stabilizes in $T$ steps, the RNN reaches a stable point $s[T]$. Its convergence is typically guaranteed by either symmetric connections with asynchronous updates or by a sufficiently small spectral radius of asymmetric connections with synchronous updates (Hopfield, 1982; Yildiz et al., 2012; Liu et al., 2026). That said, other factors, e.g. activation function, also influence the dynamical properties of RNNs (Miller & Hardt, 2019).

### 2.2 SCALING ADJACENCY MATRIX TO TUNE NETWORK DYNAMICS

Scaling the spectral radius (SR) of the adjacency matrix, the largest eigenvalue of the weight matrix, is a common method to tune the dynamics of RNN (Bai et al., 2012; Nakajima et al., 2024; Liu et al., 2026). A SR less than one yields stable and convergent dynamics. In this case, injected signals tend to decay over time, which manifests as short-term memory. A SR exceeding one can give rise to expansive or even chaotic behavior in which small perturbations are amplified. By adjusting SR, one can bias the RNN toward convergent, oscillatory, or edge-of-chaos regimes, thereby tuning computational properties, such as convergence speed or long-term memory capacity. (Jaeger & Haas, 2004; Legenstein & Maass, 2007; Miller & Hardt, 2019).

### 2.3 EQUILIBRIUM PROPAGATION

Equilibrium propagation is a learning framework initially based on energy-based models. It proceeds in two phases: a free (first) phase and a weakly clamped (second) phase. For the first phase, the RNN converges to a steady state $s^0$ under the stimulation of input alone. In the clamped phase, the network is gently nudged by the prediction error and settles to a new stable state $s^\beta$. The weight update can be simplified to a contrastive learning compatible with spiking-time-dependent plasticity (STDP) (Scellier et al., 2018). EP has been further generalized to asymmetric RNNs governed by vector field dynamics (Scellier et al., 2018). Recent work shows that asymmetry in skew-symmetric Hopfield models can improve classification performance (Høier et al., 2024).

## 2.4 Feedback Regulation and Network Structure in the Brain

Cortical areas in the brain feature dynamic regulation of feedforward and feedback connections (Felleman & Van Essen, 1991; Mejias et al., 2016; Michalareas et al., 2016; Semedo et al., 2022; Fişek et al., 2023; Wang et al., 2023). In the visual system, for instance, feedforward signals dominate immediately following the onset of external stimulus, whereas feedback signals become prominent during spontaneous activity. Dynamically regulating the strength of feedback allows the brain to optimize information integration, ensuring efficient perception and decision-making. In mammalian neocortices, information processing involves not only feedforward synaptic chains but also extensive lateral and feedback loops that interconnect disparate regions, forming a richly recursive network rather than a strictly layered structure. This topology implies short average path length between neurons and efficient information flow (Watts & Strogatz, 1998; Markov et al., 2013; Lynn & Bassett, 2019; Kulkarni & Bassett, 2025). In deep neural networks, residual connections reflect the long-range skip-layer projections observed in cortical circuits (Perich & Rajan, 2020; Holk & Mejias, 2024). They mitigate the vanishing gradient by providing skip pathways that preserve gradient (He et al., 2016).

## 3 Accelerating EP with Brain-inspired Network Properties

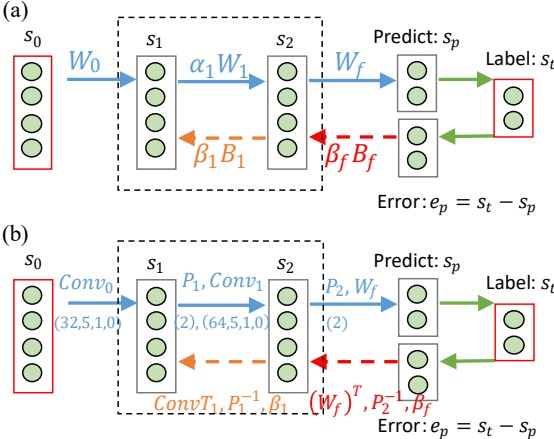

Figure 1: Illustration of feedback and feedforward regulation. (a) Layered architecture of RNN. The feedforward weights $W_i$ and feedback weights $B_i$ are rescaled by coefficients $\alpha_i$ and $\beta_i$ respectively. The dashed box encloses an RNN formed by layers $s_1$ and $s_2$ with feedforward and feedback pathways. $\beta_f$ is the nudging factor, which essentially scales the feedback strength of prediction error. (b) Embedding convolutional architecture in RNN. Convolutional parameter (32,5,1,0) is written as (channels, kernels, stride, padding). Parameter (2) in (b) denotes max-pooling with stride 2. $ConvT_i$ represents transpose convolution, the inverse process of the convolution, and $P_i^{-1}$ means max-unpooling (Ernoult et al., 2019). Model architectures and training process are given in Appendix D.

### 3.1 Prototypical setting of equilibrium propagation

Unlike the prototypical setting of equilibrium propagation (P-EP) (Ernoult et al., 2019), we separate the input and output layer from the recurrent network (Figure 1a). This separation allows the output layer to adopt the $\mathrm{SoftMax}$ activation commonly used in feedforward networks, which facilitates performance comparison (Laborieux & Zenke, 2024). For clarity, the RNN (black dashed box in Figure 1a) shown here only contains two hidden layers $s_1$ and $s_2$, but the approach applies to deeper structures (see below). The states of the RNN evolve for $T$ discrete steps until they converge. The dynamics of the whole RNN can be formulated as:

$$s^{\beta_f}[t+1] = F(s^{\beta_f}[t], b) = \rho(W \cdot s^{\beta_f}[t] + b),$$
$$b = [W_0 \cdot s_0, \quad \beta_f \cdot B_f \cdot e_p], \tag{2}$$

where $s^{\beta_f}[t]$ is the state of the RNN at time t, $\rho$ is the activation function, $W$ is the forward weight matrix of the RNN, and $b$ combines the feedforward input and the error-nudging term. Note that $s^{\beta_f} = [s_1^{\beta_f}, s_2^{\beta_f}]$. For each sample-label pair $(x, s_{tar})$, we run the free phase ($\beta_f = 0$) for $t_e$ iterations, obtain the prediction $s_p = \text{SoftMax}(W_f \cdot s_2)$, and compute the prediction error $e_p = s_{tar} - s_p$. During the clamped phase, the error nudges the RNN through the feedback weights $B_f$ and scaling coefficient $\beta_f = \beta_{f1}$ ($\beta_{f1} = 0.1$ for layered architecture and $\beta_{f1} = 0.25$ for convolutional architecture by default). The network evolves for $K$ further iterations under clamping to another state. The weights $(W_0, W_1)$ are then updated with an STDP-compatible rule:

$$\Delta W_i = ds_{i+1} \cdot (s_i^0)^\top, \quad ds_{i+1} = s_{i+1}^{\beta_{f1}} - s_{i+1}^0, \tag{3}$$

where $ds_i$ is the offset of stable point caused by the error nudging (Scellier et al., 2018). Similarly, the final weight for output is updated:

$$\Delta W_f = (s_{tar} - s_p^0) \cdot (s_2^0)^\top. \tag{4}$$

We also consider an RNN embedded with convolutional architecture in its forward paths (2 convolution layers, 2 max-pooling layers and 1 fully connected layer) shown in Figure 1b. The forward convolutional structure follows the architecture of existing convolutional neural networks (CNN) (Krizhevsky et al., 2012; Simonyan & Zisserman, 2015), in which a pooling layer is placed after the activation of the convolution layer. We transform the CNN to an RNN by adding feedback connections symmetric with the feed-forward connections (See Appendix D for the pseudocode and schematics).

## 3.2 Feedback Regulation in Layered RNN for Fast Convergence

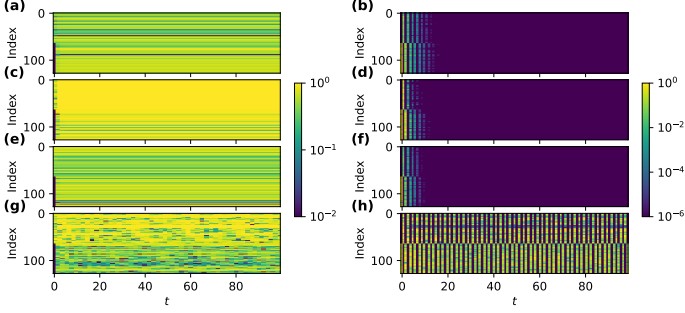

Figure 2: Convergence dynamics and speed versus feedback scaling $\beta_i$. All neurons in all hidden layers are indexed ($s_1$:0-63; $s_2$:64-127). Colors indicate neuronal activity (a,c,e,g) and changes in activity (b,d,f,h). (a) The state evolution of RNN with symmetric weights and $\beta_i = 0.1$; (b) The one-step difference of neural states in (a). (c, d) Symmetric weights with $\beta_i = 2$; (e, f) Asymmetric weights with $\beta_i = 0.1$; (g, h) Asymmetric weights with $\beta_i = 4$. In both symmetric and asymmetric feedback cases, down-scaling feedback connections tends to stabilize the network. See Figure 5d for the statistical robustness.

Although the SR can tune the RNN dynamics, scaling forward weights $W_i$ distorts forward signal propagation, which is harmful to performance (see below). Therefore, we turn to another choice, namely, scaling only the feedback strength with $\beta_i$. This coefficient scales the gradients, in the same way as the nudging factor $\beta_f$. We consider both symmetric ($B_i = (W_i)^\top$) and asymmetric ($B_i \neq (W_i)^\top$) recurrent connections in the study, and compare the results with FNNs of the same size trained by BP (feedback connections removed) or feedback alignment (FA) (Lillicrap et al., 2016) that uses random weights $B_i \neq (W_i)^\top$ to feedback the error. Note that, after scaling, the overall weight matrix $W$ of a symmetric RNN is no longer strictly symmetric. Therefore, we started from the vector field setting of EP rather then the energy-based setting in the first place. The feedforward and feedback weights are multiplied by coefficients $\alpha_i$ and $\beta_i$ respectively. Figure 2a-d shows convergence speed for different $\beta_i$. With asymmetric weights, the network can converge to a fixed point (Figure 2e, f), exhibit cyclical oscillation (Figure 2g, h), or even become chaotic. The feedback weights stay fixed during training process, which differs from EP in vector field dynamics (Scellier

et al., 2018). The pseudocode of learning procedure with a 2-hidden-layer RNN shown in Figure 1(a) is provided in Algorithm 1.

---

**Algorithm 1** EP with Feedforward and Feedback Scaling

---

**Require:** Input: $(x, s_{tar})$
**Require:** Parameters: $\theta = [W_0, W_1, W_f, B_f, B_1, \alpha_1, \beta_1, \beta_{f1}]$
**Ensure:** Updated parameters $\theta$
1: **function** FIRST-PHASE$(\theta, s_{tar})$
2:   $s_0 \leftarrow x$
3:   **for** $t = 1$ to $T$ **do**
4:    $h_1 \leftarrow W_0 \cdot s_0 + \beta_1 \cdot B_1 \cdot s_2^0$
5:    $h_2 \leftarrow \alpha_1 \cdot W_1 \cdot s_1^0$
6:    $h_p \leftarrow W_f \cdot s_2^0$
7:    $s_1^0, s_2^0, s_p^0 \leftarrow \rho(h_1), \rho(h_2), \text{SoftMax}(h_p)$
8:   **end for**
9:   $\Lambda_1 \leftarrow [s_i^0], i = 0, 1, 2, p$
10:   **return** $\Lambda_1$
11: **end function**
12: **function** SECOND-PHASE$(\theta, \Lambda_1, s_{tar})$
13:   $s_1^{\beta_{f1}}, s_2^{\beta_{f1}}, s_p^{\beta_{f1}} \leftarrow s_1^0, s_2^0, s_p^0$
14:   **for** $t = 1$ to $K$ **do**
15:    $e_p \leftarrow s_{tar} - s_p^{\beta_{f1}}$
16:    $h_1 \leftarrow W_0 \cdot s_0 + \beta_1 \cdot B_1 \cdot s_2^{\beta_{f1}}$
17:    $h_2 \leftarrow \alpha_1 \cdot W_1 \cdot s_1^{\beta_{f1}} + \beta_f \cdot B_f \cdot e_p$
18:    $h_p \leftarrow W_f \cdot s_2^{\beta_{f1}}$
19:    $s_1^{\beta_{f1}}, s_2^{\beta_{f1}}, s_p^{\beta_{f1}} \leftarrow \rho(h_1), \rho(h_2), \text{SoftMax}(h_p)$
20:   **end for**
21:   $ds_i \leftarrow s_i^{\beta_{f1}} - s_i^0, \quad i = 1, 2$
22:   $\Lambda_2 \leftarrow [ds_1, ds_2]$
23:   **return** $\Lambda_2$
24: **end function**
25: **function** UPDATING-WEIGHTS$(\theta, \Lambda_1, \Lambda_2, s_{tar})$
26:   $\Delta W_i \leftarrow ds_{i+1} \cdot (s_i^0)^\top, \quad i = 0, 1$
27:   $\Delta W_f \leftarrow (s_{tar} - s_p^0) \cdot (s_2^0)^\top$
28: **end function**

---

### 3.3 RESIDUAL CONNECTIONS TO AVOID VANISHING GRADIENTS

In our 10-hidden-layer RNN with symmetric connections, we add cross layer residual links (Figure 3a-b) and carry out ablation study on their effects in performance. The three long-range bidirectional connections bypass adjacent layers to reduce gradient decay. For RNN with asymmetric connections, we introduce skip-layer connections between non-adjacent layers with $P = 20\%$ probability, creating an RNN with arbitrary graph topologies (AGT) where any pair of layers form connections stochastically (Figure 3c) (Salvatori et al., 2022). (See Algorithm S3 in Appendix D for training detail)

## 4 EXPERIMENTS

We evaluated our RNN models on MNIST and CIFAR-10 datasets and compared the results with P-EP and BP. The MNIST dataset consists of 70,000 grayscale handwritten digit images (28×28 pixels) split into 60,000 training and 10,000 test samples. CIFAR-10 contains 60,000 RGB images (32×32 pixels) of 10 categories, divided into 50,000 training and 10,000 test samples. Pre-processing, network structures and additional training details are in Appendix D.

### 4.1 INFLUENCE OF FEEDFORWARD SCALING AND FEEDBACK SCALING

Figure 4 compares the effects of feedforward scaling $\alpha_i$ and feedback scaling $\beta_i$. In general, relative small feedback scaling ($\beta_i = 0.1$) yields high MNIST accuracy (Figure 4). In deeper RNNs, overly

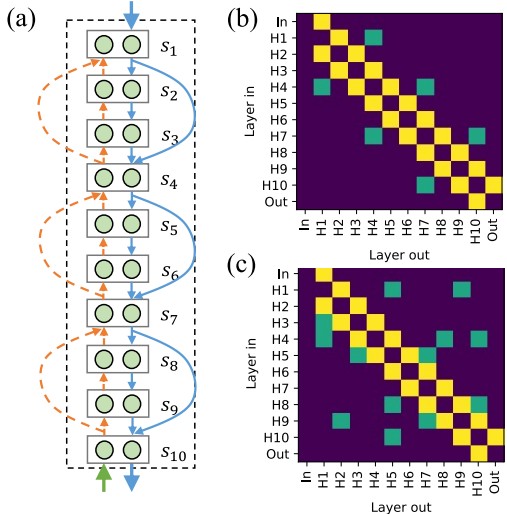

Figure 3: (a) A 10-hidden-layer RNN model with residual connections. The solid blue wires and the dashed orange wires represent forward and feedback residual connections respectively. The bidirectional connections are symmetric. (b) Adjacency matrix of (a). The blocks (green) other than the sub-diagonals indicate residual connections. (c) Adjacency matrix for an RNN with arbitrary graph topology.

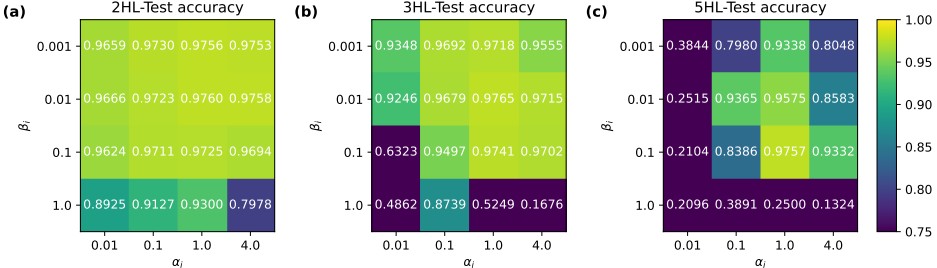

Figure 4: The influence of feedforward scaling $\alpha_i$ and feedback scaling $\beta_i$ on accuracy of MNIST classification. (a) 2 hidden layers; (b) 3 hidden layers; (c) 5 hidden layers. Each layer has 64 neurons. By default, $T = 10 \times N_{HiddenLayer}$, $K = T/2$. Each result is averaged over five repetitive experiments.

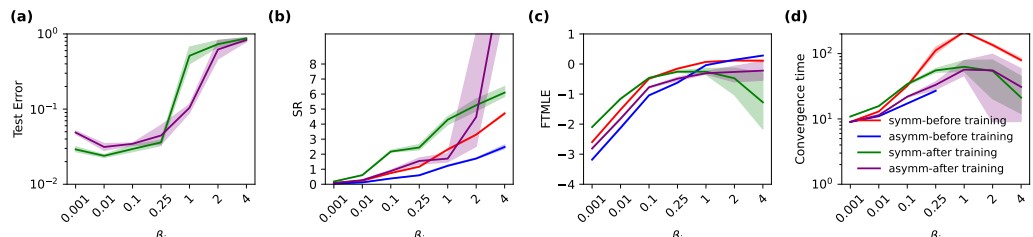

Figure 5: The test error, SR, FTMLE, and convergence time versus feedback scaling $\beta_i$. The results are obtained from a 3-hidden-layer (64 neurons per layer) model trained on MNIST dataset. Note that the network does not converge under certain conditions, resulting in missing value in d.

low feedback scaling $\beta_i$ jeopardizes the performance, which we attribute to vanishing gradients (Figure 4c, right two columns). In contrast, down-scaling the feedforward weights degrades performance, as the inference signals are weakened through the layers (see rows of Figure 4a). However, up-scaling $\alpha_i$ can also be detrimental, as this easily leads to saturation of neural state. The best performance of a 5-hidden-layer RNN is achieved without feedforward scaling $\alpha_i = 1$ and a trade-off in feedback scaling at $\beta_i = 0.1$. These results suggest that balancing the feedforward and feedback strengths is critical for better performance, not only accuracy but also speed (see Table 1).

To further investigate the influence of feedback scaling $\beta_i$, we plot the error, SR, finite time maximum Lyapunov exponent (FTMLE) (Shadden et al., 2005; Kanno & Uchida, 2014) and convergence time against feedback scaling coefficient before and after training of a 3-hidden-layer RNN on MNIST (Figure 5). It shows that larger feedback scaling $\beta_i$ decreases accuracy (Figure 5a). As expected, SR is positively correlated to $\beta_i$ (see Figure 5b), and large SR can lead to instability of an RNN indicated by the FTMLE shown in Figure 5c, which in turn explains the results in Figure 5a. In general, down-scaling the feedback ($\beta_i < 1$) reduces the convergence time of RNN, which is favorable. Note that up-scaling of feedback $\beta_i > 1$ can also decrease FTMLE and convergence time. However, this is attributed to the saturation of neural state, and will also lower the performance.

Additionally, one might suspect that the gradient signals in the lower layers are not fulfilling their intended role. In reservoir computing, where only the last layer is trained, the network can also reach high accuracy as long as the output dimension is large enough. However, this is unlikely in our case, as each layer in our network has only 64 neurons by default (other than the results in Table 1). To further confirm that the learning in lower layers is meaningful, we performed training with the weights of lower layers frozen—details of these experiments are included in Appendix C.5. The results clearly show that getting comparable results to BP requires effective training of lower layers.

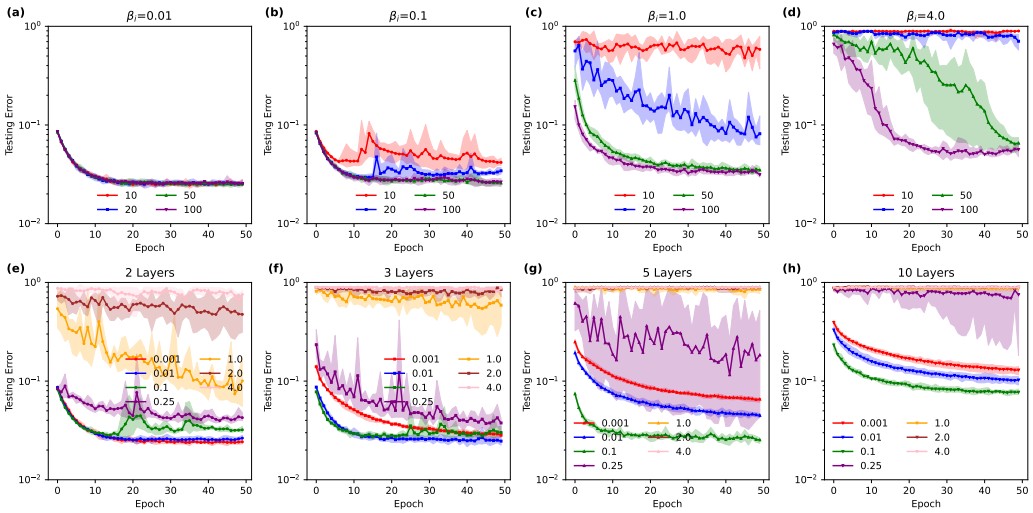

Figure 6: Test error with different hyperparameters. The curves of different T (10, 20, 50, 100) with 2 hidden layers (64 neurons per hidden layer) and (a) $\beta_i = 0.01$; (b) $\beta_i = 0.1$; (c) $\beta_i = 1$; (d) $\beta_i = 4$. The curves of different $\beta_i$ (0.001, 0.01, 0.1, 0.25, 1, 2, 4) with (e) 2 hidden layers; (f) 3 hidden layers; (g) 5 hidden layers; (h) 10 hidden layers. The shaded areas represent deviations of five repeated experiments. By default, $T = 10 \times N_{HiddenLayer}, K = T/2$. See Appendix A for more information.

## 4.2 Down-scaling Feedback Leads to Faster Convergence

Figure 6a-d plots the error versus the number of epochs with different iteration steps $T$. Under the condition of $\beta_i = 0.01$ (Figure 6a), the model with $T = 10$ and $K = 5$ works as well as the model with $T = 100$ and $K = 50$, suggesting possibility of speedup in training. Larger $\beta_i$ requires more iterations to achieve a certain level of performance (See Figure 6b, c, d). Larger $\beta_i$ means larger SR and FTMLE, thus requiring more iterations to settle the RNN as shown in Figure 2 and Figure 5(b-

Table 1: Comparison with P-EP and BP in accuracy and cost. The results of P-EP come from previous work (Ernoult et al., 2019). For BP results, we used a network with the same number of layers and number of nodes/channels. Each experiment is repeated five times, and the standard deviation is given. By default, $\beta_i = 0.01$ in our results, the feedback weights are symmetric with the feedforward weights for P-EP and Ours, and the learning rate in all layers are the same except for Ours-DLR (different learning rate), which uses varying learning rates identical to that of P-EP. For 2HL (two hidden layers) and 3HL (three hidden layers), there are 512 nodes per hidden layer. See Appendix D for more details.

| Architecture | Training approach | Testing (Training) | Epoch / Batch size -T/K | Wall Clock Time (HH:MM:SS) |
|---|---|---|---|---|
| 2HL | P-EP (sigmoid-s) | 98.05%±0.10% (99.86%) | 50/20-100/20 | 1:56:- |
| | Ours (tanh, Adam) | 98.39%±0.04% (100.00%) | 50/500-10/10 | 0:01:16 |
| | BP (tanh, Adam) | 98.26%±0.06% (100.00%) | 50/500-1/1 | 0:00:18 |
| 3HL | P-EP (sigmoid-s) | 97.99%±0.18% (99.90%) | 100/20-180/20 | 8:27:- |
| | Ours-DLR (tanh) | 97.65%±0.08% (98.93%) | 100/20-18/10 | 1:01:14 |
| | Ours (tanh) | 97.83%±0.13% (99.98%) | 100/20-18/10 | 1:01:54 |
| | Ours (tanh, Adam) | 98.36%±0.06% (100.00%) | 50/500-18/10 | 0:02:11 |
| | BP (tanh, Adam) | 98.36%±0.08% (100.00%) | 50/500-1/1 | 0:00:24 |
| Conv | P-EP (hard-sigmoid) | 98.98%±0.04% (99.46%) | 40/20-200/10 | 8:58:- |
| | Ours (hard-sigmoid) | 99.14%±0.02% (99.78%) | 40/128-20/10 | 0:12:28 |
| | BP (hard-sigmoid) | 98.93%±0.18% (99.43%) | 40/128-1/1 | 0:01:01 |

Table 2: Comparison with BP and FA and ablation study of residual connection. For layered architecture, there are 64 nodes per hidden layer and we chose $T = 10 \times N_{HiddenLayer}$, and $K = 5 \times N_{HiddenLayer}$, which guarantees saturation of accuracy at $\beta_i = 0.1$. For convolutional architectures, $\beta_i = 0.01$. By default, the Adam optimizer is used. Each experiment is repeated five times. See Appendix D for more training details.

| Architecture -connections | Training approach | MNIST-Testing (Training) | CIFAR-10-Testing (Training) |
|---|---|---|---|
| 5-symm | BP | 97.69%±0.10% (100.00%) | 49.23%±0.81% (56.72%) |
| | Ours | 97.64%±0.10% (99.98%) | 50.72%±0.17% (57.02%) |
| 5-asymm | FA | 96.44%±0.10% (98.96%) | 37.97%±2.18% (38.92%) |
| | Ours | 96.37%±0.11% (97.99%) | 45.27%±0.73% (46.79%) |
| 10-symm | BP | 97.61%±0.04% (99.93%) | 48.23%±1.26% (55.37%) |
| | Ours | 92.49%±0.32% (95.27%) | 34.90%±0.38% (34.64%) |
| | Ours-Residual | 97.49%±0.05% (99.77%) | 44.46%±0.51% (48.67%) |
| 10-asymm | FA | 94.52%±0.26% (95.54%) | 30.16%±6.12% (30.20%) |
| | Ours | 87.37%±0.49% (87.95%) | 30.37%±1.09% (29.97%) |
| | Ours-AGT | 96.87%±0.11% (99.45%) | 30.94%±4.90% (31.36%) |
| 20-symm | BP | 97.48%±0.07% (99.74%) | 47.35%±1.49% (54.59%) |
| | Ours-Residual | 95.95%±0.18% (98.20%) | 43.61%±1.17% (44.26%) |
| Conv | BP | 99.34%±0.04% (99.97%) | 75.45%±0.46% (83.61%) |
| | Ours | 99.27%±0.07% (99.78%) | 75.04%±0.51% (80.79%) |

d). Or even worse, the gradient signal is completely distorted. At $\beta_i = 4$, even T=100 fails to exceed 95% accuracy. Figure 6e-h shows that while shallow networks benefit from low $\beta_i$, deeper networks (3, 5 and 10 layers) lose accuracy. In all cases, training performance peaks at certain $\beta_i$ dependent on the network depth. Additional results are provided in Table S1 in Appendix B.

Table 1 compares our approach with P-EP, BP, and FA. Our model supersedes P-EP in training speed by at least one order of magnitude for both convolutional architecture and layered architecture. Importantly, our accuracy is comparable to BP and FA for the shallow architectures (5-hidden-layer and conv model, see also Table 2). In consideration of the improved stability (Figure 6) via feedback regulation, we anticipate that physical implementations of RNN can achieve performance on par

with BP. Additionally, for layered architecture, we also adopt the same training parameters (learning rate, batch size and epochs) as P-EP, differing only in feedback scaling ('ours-DLR' in Table 1). The results present clear evidence of speedup, which mainly stems from the reduced number of iterations required for convergence.

### 4.3 DOWN-SCALED FEEDBACK COORDINATES PLASTICITY OF DIFFERENT LAYERS

It is hypothesized that the brain requires different plasticity in different areas due to their varying functional roles (Atallah et al., 2004; Lowet et al., 2020). The variability in plasticity can be realized explicitly by adjusting learning rates or implicitly by modulating the intensity of gradient. Previous work postulated that EP with weak feedback necessitates learning rates differing by orders of magnitude across layers (Scellier & Bengio, 2017). Here, we found that due to gradient differences across different layers induced by weak feedback, a 3-hidden-layer RNN at $\beta_i = 0.01$ (Table 1, 'ours (tanh)') learns well with a uniform learning rate. This result suggests that the feedback scaling alone is able to regulate gradient strength of different layers, pointing to another possible mechanism to coordinate plasticity.

### 4.4 RESIDUAL CONNECTIONS OVERCOME THE GRADIENT VANISHING IN DEEP RNNS

Weak feedback exacerbates vanishing gradient in deeper layered RNN (Figures S5–S6 in Appendix B). Adding residual connections restores gradient flow (Figure S7 in Appendix B). As a result, a 10-hidden-layer network sees substantial performance gains (Table 2), 5% increase in accuracy for MNIST and 9% for CIFAR-10. Even 20-hidden-layer model can be trained. As shown in Table 2, without residual connections, an asymmetric RNN trained by EP falls short of FA in accuracy, but arbitrary residual links surpass the accuracy of FA for the MNIST classification (See ablation study on connection probability in Appendix B.). For more complex dataset CIFAR-10, the 10-hidden-layer asymmetric model with residual random feedback connections achieves accuracy nearly 14% below the symmetric model. A possible reason is that the gradient signal through multiple random fixed feedback connections becomes too distorted by error to coordinate the forward weight learning.

## 5 DISCUSSION

We have applied the feedback scaling to RNN to speed up the convergence and to accelerate training with EP with negligible overhead. To counteract the vanishing gradient in deep architectures, we have added residual connections to non-adjacent layers of deep RNNs, partly restoring classification performance. In principle, the residual connections make credit assignment pathways shorter (Veit et al., 2016). The training exhibits remarkable resilience to noise on weight and neural state. Our structural modification is compatible with other algorithmic speed-ups (Scellier et al., 2023), thereby expanding the design space for efficient EP implementations.

Recent work on credit assignment in brain-inspired networks, e.g. adjoint propagation (Liu et al., 2026), partitions a large network into local RNNs with random internal connections of low SR for fast convergence and dynamic resource allocation, yielding speed and accuracy similar to this work. This work, however, adopts the feedback scaling to solve the stability issue and accelerate convergence of EP.

Weak feedback is often considered in biologically plausible learning algorithms (Sacramento et al., 2018; Haider et al., 2021; Meulemans et al., 2021). It has been shown that contrastive Hebbian learning with weak feedback approximates backpropagation while converging quickly (Xie & Seung, 2003). More recently, local representation alignment (LRA) likewise employed weak feedback (Ororbia et al., 2023) and skip connections from the output to deep layers for efficient training. The EP framework also approximates BP (Scellier & Bengio, 2017; Millidge et al., 2023), but under the weak clamping condition (weak supervision) (Laborieux et al., 2021; Millidge et al., 2023). We have shown that, at the infinitesimal inference limit, namely weak supervision and weak feedback (Millidge et al., 2023), EP is equivalent to LRA and BP (Appendix C). In other words, the dynamics of FRE-RNN is more like the feedforward neural network due to its weak feedback.

However, there are still a few limitations to our approaches for large-scale neural networks that underpin artificial intelligence. For complex datasets like CIFAR-10, there exists a notable performance gap compared to BP, using deep fully connected neural networks. We attribute this gap to the inaccurate approximation to the true gradient as computed by BP (See Appendix C.4). Therefore, although EP can be extended to deep fully connected network (20-hidden-layers) and shallow CNNs, its applicability for deep CNN remains to be explored. For deep architectures with asymmetric connections, the accuracy decreases faster with increasing depth due to the inaccurate random error feedback. More in-depth investigation on residual connection topology is required to scale up the methodology to large scale deep architectures. Besides, the hyperparameters are optimized empirically. We find a feedback scaling in the range of 0.01-0.1 is favorable for shallow networks (less than 4 layers) and 0.1-0.25 for deeper architectures. Finding a general way to determine these parameters is still on-going. Additionally, existing research on EP converging naturally continues to focus primarily on static-input settings (Laborieux et al., 2021; Ernoult et al., 2019; Laborieux & Zenke, 2024). Extending naturally converging RNN trained by EP to sequence tasks remains a challenge.

From a neurobiological perspective, residual connections, particularly the randomly generated arbitrary graph topologies, yield cortex-like connectivity patterns in the brain. The feedback-regulated residual RNNs equip the biologically plausible learning framework, EP, with biologically plausible network architecture. Although it currently runs on GPUs, it can exploit the natural convergence of physical RNNs and facilitate efficient learning and inference on dedicated neuromorphic hardware.

ACKNOWLEDGEMENTS

This work was supported by the National Key R&D Program of China (Grant No. 2024YFA1208804). Additional financial support from the University of Science and Technology of China and the Chinese Academy of Sciences is also gratefully acknowledged.

CODE AVAILABILITY

The code used in this work is available at `https://github.com/Zero0Hero/FRE-RNN-EP`.

REPRODUCIBILITY STATEMENT

The code necessary to reproduce the main results is provided as Jupyter Notebooks in the Supplementary Materials. Researchers can directly run them to reproduce the results. Further details on data pre-processing and training process are available within the provided code and in Appendix D.

THE USE OF LARGE LANGUAGE MODELS (LLMS)

In the preparation of this work, the authors used GPT-5 and DeepSeek solely for the purpose of polishing and improving the linguistic fluency and readability of the text. This includes tasks such as correcting grammar and rephrasing sentences. After using the model, the authors have reviewed and edited all content extensively and take full responsibility for all ideas, claims, and the final language presented in this paper.

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

# A    THE DYNAMICS OF THE RNN

We quantify the convergence property of the recurrent neural network (RNN) with maximum Lyapunov exponent (MLE) (Wolf et al., 1985), and finite time maximum Lyapunov exponent (FTMLE) (Kanno & Uchida, 2014). When the MLE/FTMLE is large, the RNN converges slow or even not at all. To compute MLE and FTMLE, we first initialize a random perturbation vector $\delta_0$. Then we record the sequence of states $s^0[t]$ with $t = 0, 1, 2, \ldots, T_e - 1$ corresponding to the last sample of a training set (see Figure 2 in the main text), and run the following steps:

1. Normalize perturbation vectors to unit length:

$$\delta_t \leftarrow \frac{\delta_t}{\|\delta_t\|}$$

2. Calculate the Jacobian matrix:

$$J(s^0[t]) = \frac{\partial F(s^0[t], b)}{\partial s^0[t]}$$

3. Update the perturbation:

$$\delta_{t+1} = J(s^0[t]) \cdot \delta_t$$

4. Record

$$r_i = \ln \|\delta_{t+1}\|$$

The maximum Lyapunov exponent is computed as $\lambda_{\max} = \frac{1}{T_e} \sum_{t=0}^{T_e - 1} r_i$ for a sufficiently large $T_e$ (default $T_e = 500$). The results at any $T < T_e$ are the FTMLE.

Figure S1–S2 show the FTMLE, MLE, training accuracy and test accuracy versus epochs of different models. In all cases, smaller $\beta_i$ usually yields smaller (FT)MLE, whereas larger $\beta_i$ do not always lead to larger (FT)MLE because the activation function saturates. The saturation diminishes perturbation.

For 2-hidden-layer RNN, smaller feedback scaling $\beta_i$ yields steady training progress and better accuracy. Figure S3 plots the FTMLE and test accuracy against feedback scaling for different numbers of hidden layers. It shows that smaller $\beta_i$ is favorable for shallow networks, because the RNN is easier to converge (indicated by FTMLE). But for deeper networks (5-hidden-layer or more), smaller $\beta_i$ degrades performance because of vanishing gradient.

Further comparison between our FRE-RNN that incorporates convolutional structure with previous work (Ernoult et al., 2019) are also plotted in Figure S4. These results suggest that small feedback scaling ($\beta_i = 0.01$) leads to a smoother training process.

# B    GRADIENT VANISHING AND THE RESIDUAL CONNECTIONS

Figure S5 and S6 plot the error of each neuron versus epoch at different $\beta_i$. For a 2-hidden-layer RNN, the best performance is obtained at $\beta_i = 0.001$. In this situation, the error of the first hidden layer is at least two orders of magnitude less than the second hidden layer. At $\beta_i = 2$, the error also decreases from higher (high index neurons, closer to output layer) to lower layers, which is attributed to the saturation of the activation function. In general, the training progresses more steadily for smaller $\beta_i$ despite the vanishing gradient, which also applies to deeper networks (up to 10-hidden-layer).

To eliminate the vanishing gradient in EP, direct feedback from the higher layers or local amplification (with higher learning rate) is unavoidable (Nøkland, 2016; Ororbia et al., 2023). Figure S7 shows the effect of residual connections. $\beta_i = 0.1$ yield the best accuracy $97.5\%$, due to the balance between gradient flow and convergence.

Figure S8 shows the testing accuracy varies with the connection probability $P$ of AGT with 10 hidden layers. Except for the connections in layered model, the connection between any two hidden layers is generated with probability $P$, i.e., we first use $P$ to decide if the connections between any two layers will be established. As P increases, the accuracy rises first, peaks at 0.2 and decreases around 1. However, the reason behind is yet to be explored.

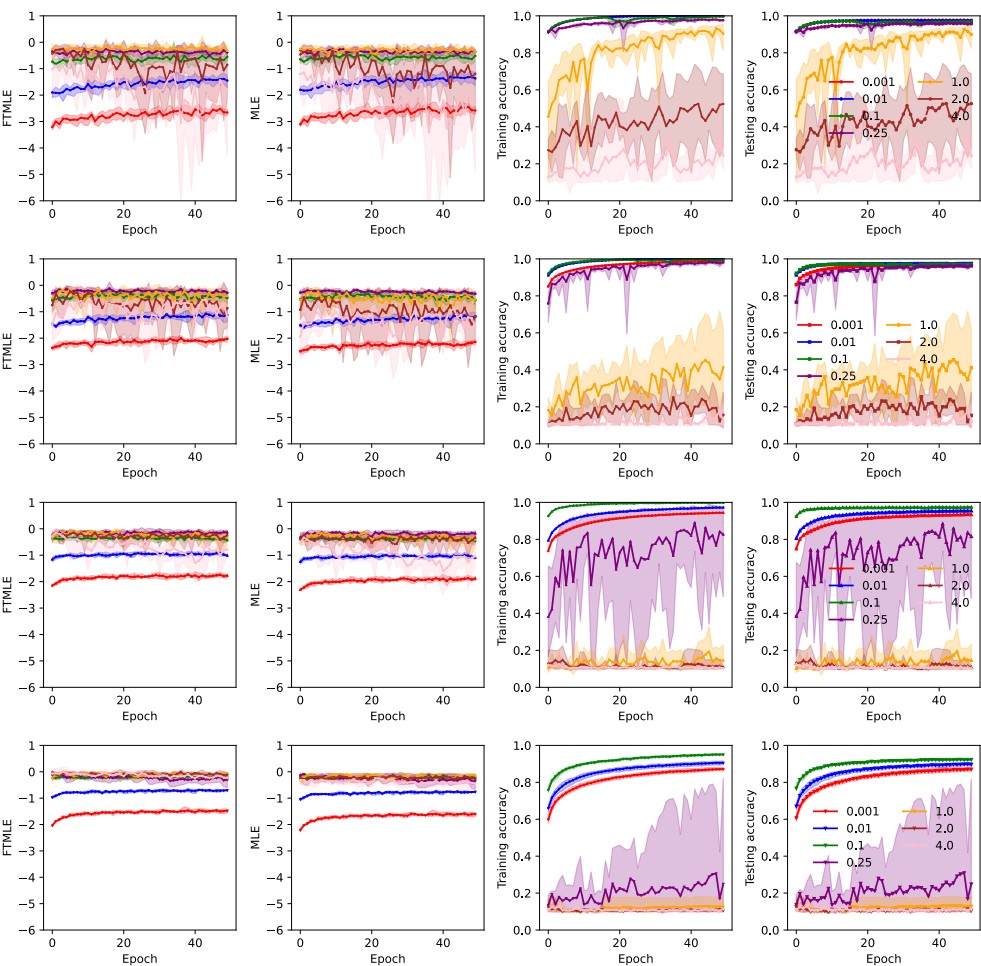

Figure S1: The FTMLE, MLE, training accuracy and testing accuracy of symmetric RNNs versus epochs with different feedback scaling $\beta_i$ (legend). First row: 2 hidden layers; Second row: 3 hidden layers; Third row: 5 hidden layers; Fourth row: 10 hidden layers. The activation is tanh. Each case is repeated 5 times.

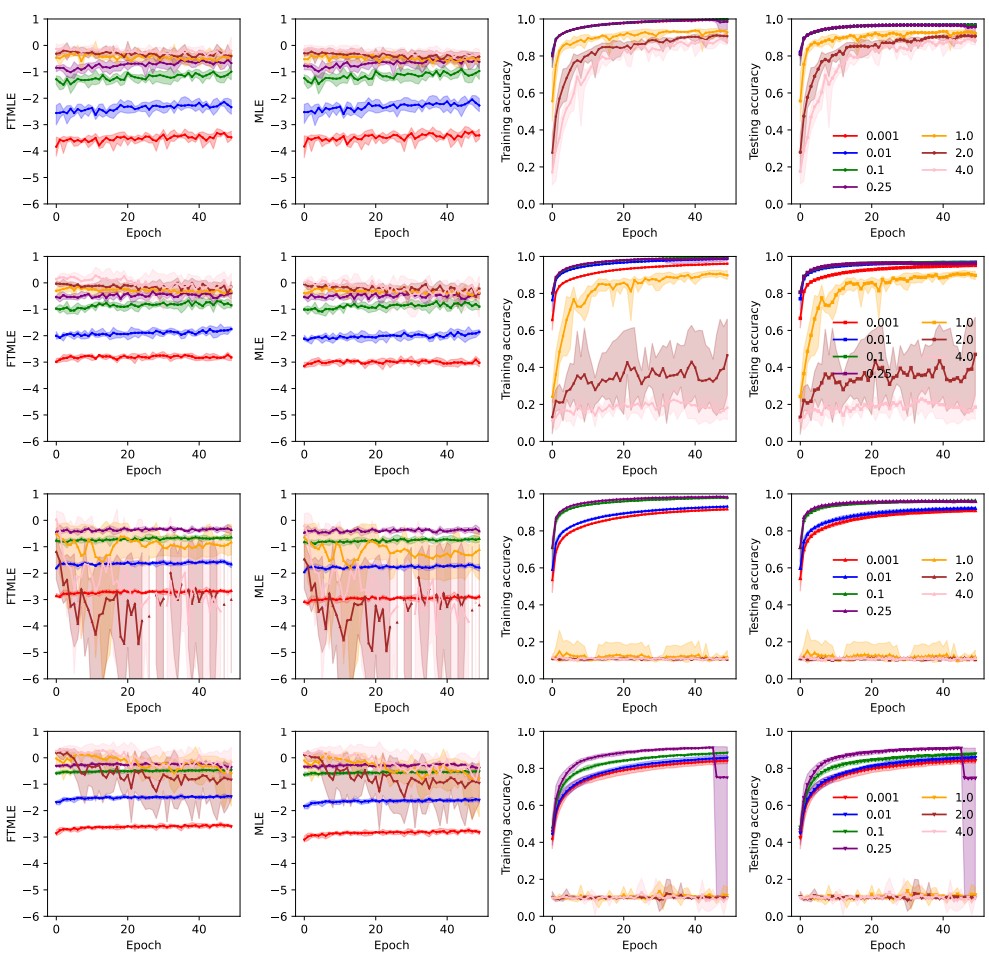

Figure S2: The FTMLE, MLE, training accuracy and testing accuracy of asymmetric RNNs versus epochs with different feedback scaling $\beta_i$ (legend). First row: 2 hidden layers; Second row: 3 hidden layers; Third row: 5 hidden layers; Fourth row: 10 hidden layers. The activation is tanh. Each case is repeated 5 times.

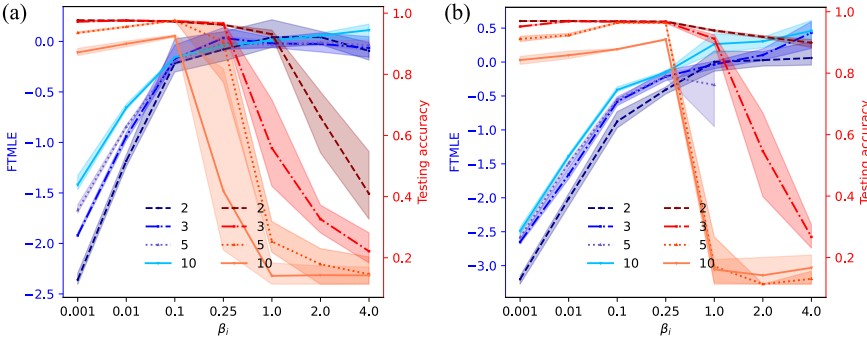

Figure S3: The FTMLE and testing accuracy versus feedback scaling $\beta_i$ with different numbers of hidden layers. (a) Symmetry weights; (b) Asymmetric weights. The FTMLE and testing accuracy given here correspond to their maxima in all epochs. Note that the 5-hidden-layer asymmetric RNN with large $\beta_i$ diverged and resulted in missing data points in (b). Each case is repeated 5 times.

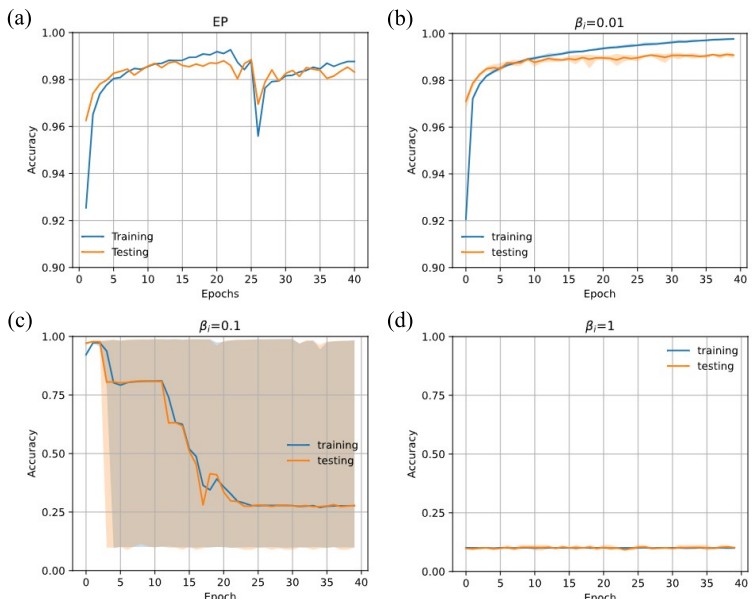

Figure S4: Comparison of RNN embedded with convolutional structure on the MNIST between P-EP (a) (Ernoult et al., 2019) and our approach at different $\beta_i$ (b-d). We used the same parameters as the EP reference (Ernoult et al., 2019).

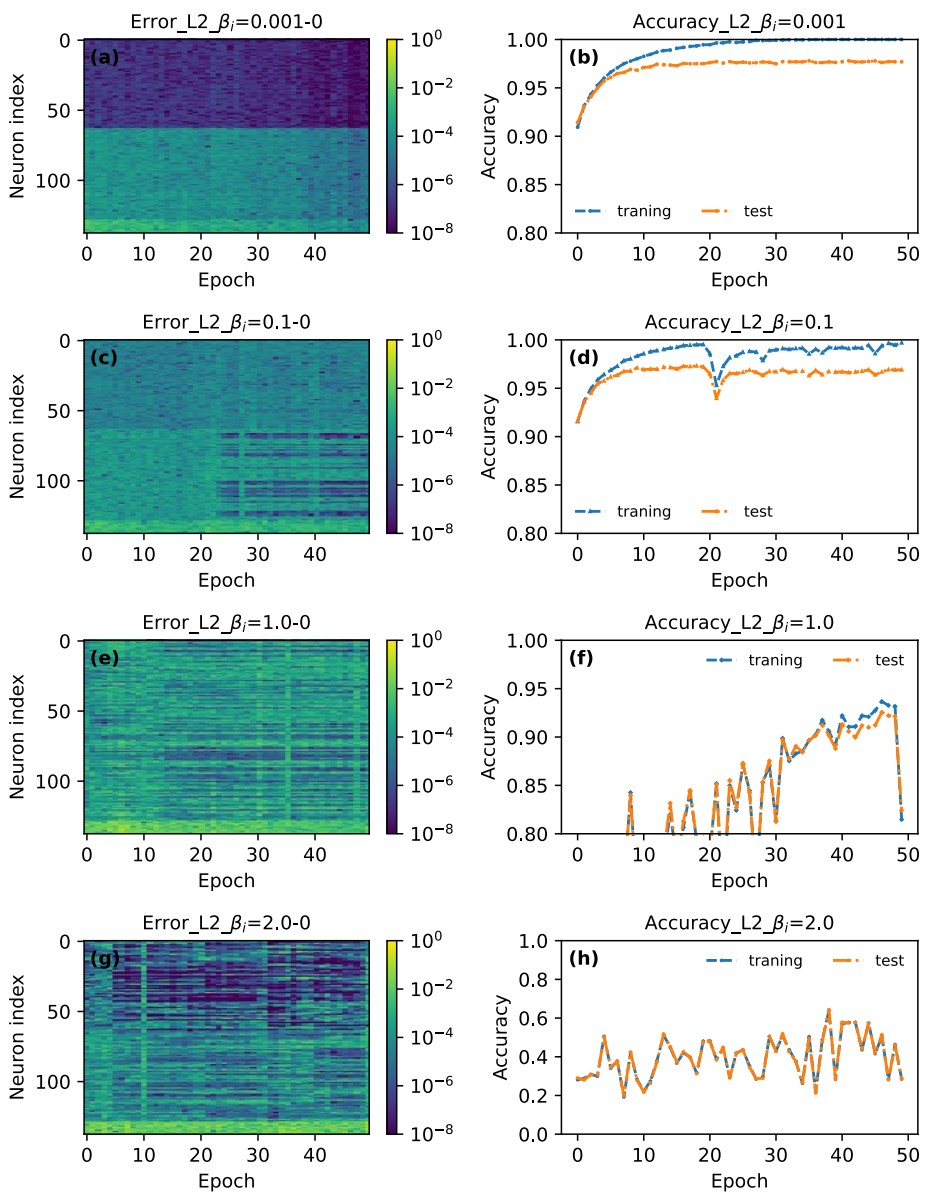

Figure S5: For 2-hidden-layer RNN, the mean error of each neuron in the last batch and testing accuracy versus epochs at different $\beta_i$. All neurons in the hidden layers and the output layer are indexed from the input to the output layer.

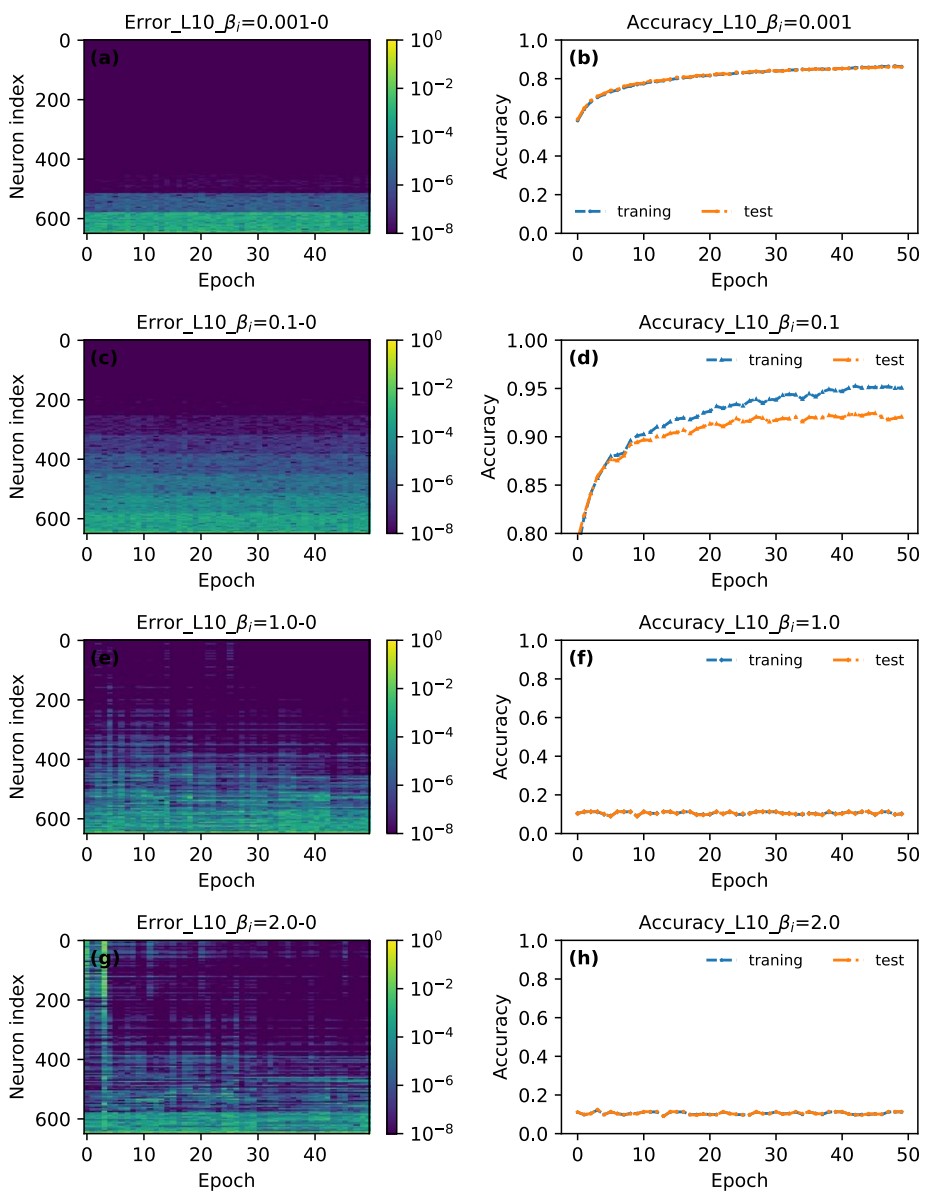

Figure S6: For the 10-hidden-layer model, the mean error of each neuron in the last batch and testing accuracy versus epochs at different $\beta_i$. All neurons in the hidden layers and the output layer are indexed from the input to the output layer.

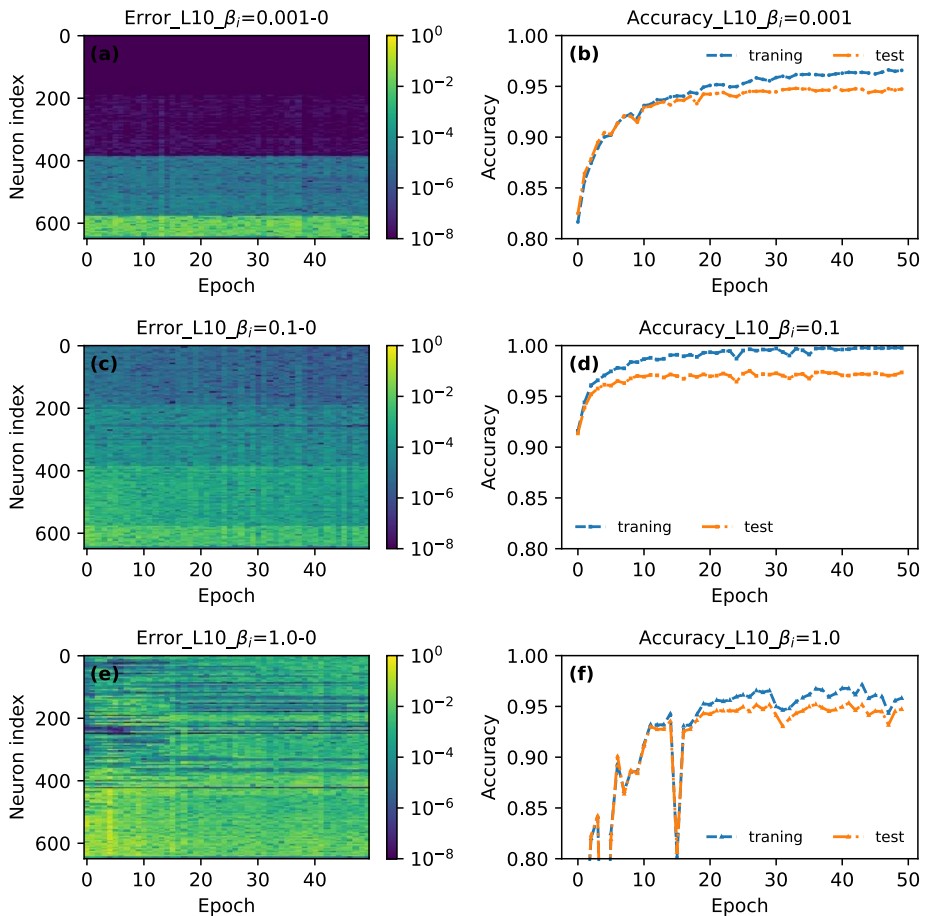

Figure S7: For the 10-hidden-layer model with residual connections, the mean error of each neuron in the last batch and testing accuracy versus epochs at different $\beta_i$. All neurons in the hidden layers and the output layer are indexed from the input to the output layer.

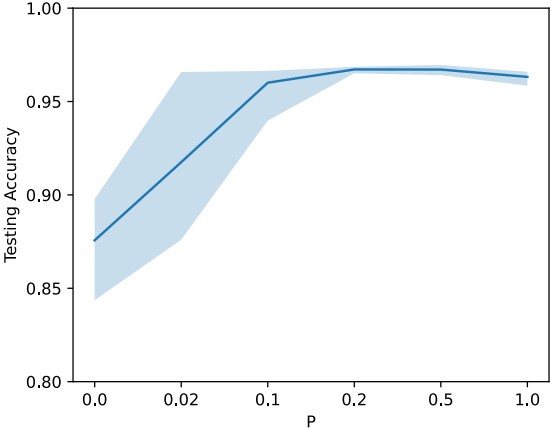

Figure S8: The testing accuracy on MNIST varies with the connection probability $P$ of AGT with 10 hidden layers. The experiments are repeated 5 times.

Table S1: Testing accuracy (mean of 5 repeated experiments) with different feedback scaling $\beta_i$. By default, $T = 10 \times N_{HiddenLayer}$, $K = 5 \times N_{HiddenLayer}$. Each hidden layer has 64 nodes.

| Architecture-connections | $\beta_i = 0.001$ | $\beta_i = 0.01$ | $\beta_i = 0.1$ | $\beta_i = 0.25$ | $\beta_i = 1$ | $\beta_i = 2$ | $\beta_i = 4$ |
|---|---|---|---|---|---|---|---|
| 2HL-symm | 97.69% | 97.57% | 97.25% | 96.22% | 93.12% | 66.04% | 40.92% |
| 3HL-symm | 97.22% | 97.64% | 97.41% | 96.60% | 55.86% | 32.64% | 22.11% |
| 5HL-symm | 93.54% | 95.54% | 97.60% | 90.63% | 25.31% | 17.88% | 14.61% |
| 10HL-symm | 87.15% | 89.99% | 92.54% | 41.84% | 14.07% | 14.30% | 14.23% |
| 10HL-Residual-symm | – | 97.52% | 97.46% | – | 95.51% | – | – |
| conv-symm | – | 99.15% | 98.71% | – | 11.35% | – | – |
| 2HL-asymm | 96.96% | 96.97% | 96.88% | 96.79% | 93.88% | 91.81% | 89.91% |
| 3HL-asymm | 95.17% | 96.91% | 96.76% | 96.66% | 91.21% | 54.65% | 26.72% |
| 5HL-asymm | 91.14% | 92.34% | 96.41% | 96.35% | 17.15% | 11.35% | 13.07% |
| 10HL-asymm | 84.27% | 85.83% | 87.79% | 90.97% | 16.13% | 14.21% | 16.67% |
| 10HL-AGT-asymm | – | 96.37% | 96.75% | – | 33.31% | – | – |

## C  EQUIVALENCE WITH EP AND BP UNDER THE CONDITION OF INFINITESIMAL INFERENCE LIMIT

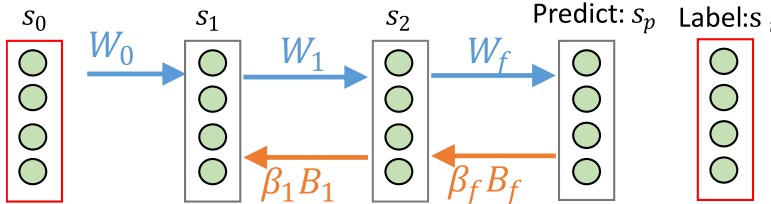

Figure S9: A layered network model used to illustrate the process of backpropagation (BP), local representation alignment (LRA), and EP. Note that the final prediction layer $\cdot_p$ corresponds to the third layer with subindex $\cdot_3$. For LRA, we use $\beta_{LRA}$ instead of $\beta_1$ and $\beta_f$. For BP, the feedback (orange) paths are absent.

In this section, we will use the infinitesimal inference limit (Millidge et al., 2023) to derive the equivalence of EP with LRA and BP.

### C.1  BACKPROPAGATION

When we remove the feedback connection of a 2-hidden-layer RNN shown in Figure S9, a feedforward network is left and can be trained with BP. The forward process of BP is described by:

$$
\begin{aligned}
s_1 &= \rho(h_1), h_1 = W_0 \cdot s_0, \\
s_2 &= \rho(h_2), h_2 = W_1 \cdot s_1, \\
s_p &= h_p, h_p = W_f \cdot s_2.
\end{aligned} \tag{S1}
$$

Defining a loss $L_{\text{BP}} = \frac{1}{2}(s_p - s_{tar})^2$, the weights adjust according to the gradient of the loss. Taking $\Delta W_0$ as an example:

$$
\begin{aligned}
\Delta W_0 &= -\frac{\partial L_{\text{BP}}}{\partial W_0} \\
&= -\rho'(h_1) \odot W_1^\top \cdot \left( \rho'(h_2) \odot W_f^\top \cdot (s_p - s_{tar}) \right) \cdot (s_0)^\top,
\end{aligned} \tag{S2}
$$

where "$\odot$" means Hadamard product (element-wise product), "$\cdot$" means scalar or matrix multiplication. For two vectors/matrices, "$\odot$" requires identical dimensions and computes element-wise products. Broadcasting rules may apply (e.g., a column vector $v_{m \times 1} \odot A_{m \times n}$ scales each column of $A$ by $v$).

## C.2 LOCAL REPRESENTATION ALIGNMENT

LRA is an alternative training method following the principle of discrepancy reduction (Ororbia et al., 2017; Ororbia & Mali, 2019). It can be divided into two phases: 1) the network runs the forward process, producing latent representations of the input samples. 2) the weights adjust in the direction of reducing the mismatch between current latent representations and target representations in each layer.

The forward process is the same as BP:

$$
\begin{aligned}
s_1^0 &= \rho(h_1^0), \quad h_1^0 = W_0 \cdot s_0, \\
s_2^0 &= \rho(h_2^0), \quad h_2^0 = W_1 \cdot s_1^0, \\
s_p^0 &= h_p^0, \quad h_p^0 = W_f \cdot s_2^0.
\end{aligned} \tag{S3}
$$

where $s_i^0$ are interpreted as the latent representations. The prediction error is $e_p = s_{tar} - s_p^0$. Then we can get the target representations of the second hidden layer:

$$
s_2^{\beta_{\text{LRA}}} = \rho(h_2^{\beta_{\text{LRA}}}), \quad h_2^{\beta_{\text{LRA}}} = W_1 \cdot s_1^0 + \beta_{\text{LRA}} \cdot B_f \cdot e_p, \tag{S4}
$$

The same goes for the first hidden layer:

$$
s_1^{\beta_{\text{LRA}}} = \rho(h_1^{\beta_{\text{LRA}}}), \quad h_1^{\beta_{\text{LRA}}} = W_1 \cdot s_0 + \beta_{\text{LRA}} \cdot B_1 \cdot e_2, \quad e_2 = s_2^{\beta_{\text{LRA}}} - s_2^0, \tag{S5}
$$

LRA defines the loss as the total discrepancy between latent representations and target representations:

$$
L_{\text{LRA}} = \sum_{i=1}^{L} k_i L_i(s_i^0, s_i^{\beta_{\text{LRA}}}) = \sum_{i=1}^{L} \frac{1}{2}(s_i^0 - s_i^{\beta_{\text{LRA}}})^2, \tag{S6}
$$

The weight $W_i$ adjusts according to the local mismatch between $s_{i+1}^0$ and $s_{i+1}^{\beta_{\text{LRA}}}$:

$$
\begin{aligned}
\Delta W_i &= -\frac{\partial k_i L_i(s_{i+1}^0, s_{i+1}^{\beta_{\text{LRA}}})}{\partial W_i} \\
&= (s_{i+1}^{\beta_{\text{LRA}}} - s_{i+1}^0) \odot f'(h_{i+1}^0) \cdot (s_i^0)^\top \\
&\approx (s_{i+1}^{\beta_{\text{LRA}}} - s_{i+1}^0) \cdot (s_i^0)^\top,
\end{aligned} \tag{S7}
$$

where the derivative of the activation function is omitted in the last row, a useful practice common in LRA (Melchior & Wiskott, 2019; Ororbia & Mali, 2019; Ororbia et al., 2023). When $\beta_{\text{LRA}} \to 0$, $s_i^{\beta_{\text{LRA}}} \to s_i^0$ and $h_i^{\beta_{\text{LRA}}} \to h_i^0$, then

$$
\begin{aligned}
e_i &= s_i^{\beta_{\text{LRA}}} - s_i^0 = \rho(h_i^{\beta_{\text{LRA}}}) - \rho(h_i^0) \\
&= \rho(h_i^0 + \beta_{\text{LRA}} \cdot B_i \cdot e_{i+1}) - \rho(h_i^0) \\
&\approx [\rho(h_i^0) + \rho'(h_i^0) \odot (\beta_{\text{LRA}} \cdot B_i \cdot e_{i+1}) - \rho(h_i^0)]_{\beta_{\text{LRA}} \to 0}, \\
&= \rho'(h_i^0) \odot (\beta_{\text{LRA}} \cdot B_i \cdot e_{i+1})
\end{aligned} \tag{S8}
$$

The approximation in Equation S8 is based on a first-order Taylor expansion of $\rho(h_i^0 + \Delta h)$ around $h_i^0$, where $\Delta h = \beta_{\text{LRA}} \cdot B_i \cdot e_{i+1}$. For a small perturbation $\Delta h \to 0$, the Taylor expansion gives:

$$
\rho(h_i^0 + \Delta h) = \rho(h_i^0) + \rho'(h_i^0) \cdot \Delta h + O(\Delta h^2), \tag{S9}
$$

When $\beta_{\text{LRA}} \to 0$, higher order terms $O(\Delta h^2)$ are negligible, leaving only the linear terms. We arrive at the last row after canceling out $\rho(h_i^0)$. There we can express the weight adjustments as

$$
\begin{aligned}
\Delta W_0 &= e_1 \cdot (s_0^0)^\top \\
&= \left[ \rho'(h_1^0) \odot \left( \beta_{\text{LRA}} \cdot B_1 \cdot \left( \rho'(h_2^0) \odot (\beta_{\text{LRA}} \cdot B_f \cdot (s_{tar} - s_p)) \right) \right) \right] \cdot (s_0)^\top \Big]_{B_i=(W_i)^\top} \\
&= -\beta_{\text{LRA}} \cdot \beta_{\text{LRA}} \cdot \rho'(h_1^0) \odot W_1^\top \cdot \left( \rho'(h_2^0) \odot W_f^\top \cdot (s_p - s_{tar}) \right) \cdot (s_0)^\top,
\end{aligned} \tag{S10}
$$

which is the same as BP (Equation S2) except for a constant. Thus, LRA at weak feedback limit approximates BP. An LRA algorithm for a 2-hidden-layer network is described in Algorithm S1. The feedback weights in LRA need not be learned here, but can be kept symmetric with the feedforward weights.

## C.3   EQUILIBRIUM PROPAGATION

We can also formulate EP in terms of discrepancy reduction. In EP (Algorithm 1 in the main text), the network states evolve as follows ($\beta = 0$ for the first phase and $\beta = \beta_f$ for the second phase):

$$
\begin{aligned}
h_1^\beta &= W_0 \cdot s_0^\beta + \beta_1 \cdot B_1 \cdot s_2^\beta, \\
h_2^\beta &= W_1 \cdot s_1^\beta + \beta_f \cdot B_f \cdot e_p, \\
h_p^\beta &= W_f \cdot s_2^\beta, \\
s_1^\beta, s_2^\beta, s_p^\beta &= \rho(h_1^\beta), \rho(h_2^\beta), h_p^\beta,
\end{aligned}
\tag{S11}
$$

where $e_p = s_{tar} - s_p^0$ is the predicting error. The network converges to final states $h_1^0, h_2^0, s_1^0, s_2^0$ in the free phase. The error of $s_2$ neurons can be described by:

$$
\begin{aligned}
ds_2 &= [\rho(h_2^{\beta_f})]_{\beta_f \to 0} - [\rho(h_2^0)]_{\beta_f = 0} \\
&\approx \rho'(h_2^0) \odot (\beta_f \cdot B_f \cdot e_p),
\end{aligned}
\tag{S12}
$$

where only the first-order infinitesimal term is retained as $\beta_1 \to 0$. The same goes for the first hidden layer:

$$
\begin{aligned}
ds_1 &= [\rho(h_1^{\beta_f})]_{\beta_f \to 0} - [\rho(h_1^0)]_{\beta_f = 0} \\
&\approx \rho'(h_1^0) \odot (\beta_1 \cdot B_1 \cdot (\rho'(h_2^0) \odot (\beta_f \cdot B_f \cdot e_p))),
\end{aligned}
\tag{S13}
$$

The weight $W_0$ can be updated by:

$$
\Delta W_0 = \frac{ds_1 \cdot (s_0^0)^\top}{\beta_1 \cdot \beta_f} = \rho'(h_1^0) \odot B_1 \cdot (\rho'(h_2^0) \odot B_f \cdot e_p) \cdot (s_0^0)^\top,
\tag{S14}
$$

With $B_i = W_i^\top$,

$$
ds_1 = \beta_f \cdot \beta_1 \cdot \rho'(h_1^0) \odot W_1^\top \cdot (\rho'(h_2^0) \odot W_f^\top \cdot -(s_p - s_{tar})),
\tag{S15}
$$

$$
\Delta W_0 = \frac{ds_1 \cdot (s_0^0)^\top}{\beta_1 \cdot \beta_f} = -\rho'(h_1^0) \odot W_1^\top \cdot \left(\rho'(h_2^0) \odot W_f^\top \cdot (s_p - s_{tar})\right) \cdot (s_0^0)^\top.
\tag{S16}
$$

Note that compared with the weight update in the main text, $1/(\beta_1 \cdot \beta_f)$ is added to recover a gradient amplitude similar to BP. Further, if we assume that the high-order infinitesimal in the first phase can be omitted, the dynamics of RNN is governed by:

$$
\begin{aligned}
s_1^0 &= \rho(h_1^\beta), \quad h_1^0 = [W_0 \cdot s_0 + \beta_1 \cdot B_1 \cdot s_2^0]_{\beta_1 \to 0} \approx W_0 \cdot s_0, \tag{S17} \\
s_2^0 &= \rho(h_2^0), \quad h_2^0 = [W_1 \cdot s_1^0 + \beta_f \cdot B_f \cdot e_p]_{\beta_1 \to 0, \beta_f = 0} \approx W_1 \cdot s_1^0, \tag{S18} \\
s_p^0 &= h_p^0, \quad h_p^0 = W_f \cdot s_2^0. \tag{S19}
\end{aligned}
$$

The information flow of RNN degenerates into that of a feedforward network. This does not affect the error information $ds_i$, thus Equation S16 approximates Equation S2 for BP. Meanwhile, it resembles LRA with low $\beta_{\text{LRA}}$, which turns explicit error into implicit error. Hitherto, we have shown that although the errors are obtained differently in EP, LRA, and BP, they are equivalent under the assumption of weak supervision and weak feedback.

---

**Algorithm S1** Local Representation Alignment (LRA)

---

**Input:** $(x, s_{tar})$
**Parameter:** $\theta = [W_0, W_1, W_2, B_2, B_1, \beta_{\text{LRA}}]$
**Output:** $\theta$
1: **function** FORWARD$(\theta, x)$
2:      $s_0 \leftarrow x$
3:      $s_1^0 \leftarrow \rho(h_1), \quad h_1 \leftarrow W_0 \cdot s_0$
4:      $s_2^0 \leftarrow \rho(h_2), \quad h_2 \leftarrow W_1 \cdot s_1^0$
5:      $s_p^0 \leftarrow W_f \cdot s_2^0$
6:      $\Lambda_1 \leftarrow [s_i^0], \, i = 0, 1, 2, p$
7:      **return** $\Lambda_1$
8: **end function**

9: **function** FEEDBACK$(\theta, \Lambda_1, s_{tar})$
10:      $e_p \leftarrow s_{tar} - s_p^0$
11:      $s_2^{\beta_{\text{LRA}}} \leftarrow \rho(h_2), \quad h_2 \leftarrow W_1 \cdot s_1^0 + \beta_{\text{LRA}} \cdot B_f \cdot e_p$
12:      $e_2 \leftarrow s_2^{\beta_{\text{LRA}}} - s_2^0$
13:      $s_1^{\beta_{\text{LRA}}} \leftarrow \rho(h_1), \quad h_1 \leftarrow W_0 \cdot s_0 + \beta_{\text{LRA}} \cdot B_1 \cdot e_2$
14:      $e_1 \leftarrow s_1^{\beta_{\text{LRA}}} - s_1^0$
15:      $\Lambda_2 \leftarrow [e_1, e_2, e_p]$
16:      **return** $\Lambda_2$
17: **end function**

18: **function** UPDATING-WEIGHTS$(\theta, \Lambda_1, \Lambda_2)$
19:      $\Delta W_i \leftarrow e_{i+1} \cdot (s_i^0)^T, \quad i = 0, 1$
20:      $\Delta W_f \leftarrow e_p \cdot (s_2^0)^T$
21: **end function**

---

## C.4 EXPERIMENTS FOR EQUIVALENCE WITH EP AND BP

Prior works have shown that EP can be equalized to BPTT in specific conditions and can achieve comparable performance (Ernoult et al., 2019; Laborieux et al., 2021). As discussed in the previous section, although the overall architecture forms an RNN, the network behaves similarly to a feedforward model due to weak feedback connections.

To experimentally show the equivalence of EP and BP, we can further compare our model with FNN with same feedforward weights trained by BP. We mainly compare cosine similarity of states, bias gradients and weight gradients for the first batch (batch size is 200) as given in Figure S10. Figure S10(a-c) shows similarity under the conditions of $\beta_i = 1$ with different iterations. For the the bias gradients, i.e., $ds_i$, the cosine similarity declines rapidly, indicating no similarity between our model and BP. With weak feedback $\beta_i = 0.1$, as shown in Figure S10(d-f), the similarity of states approaches 1 and the similarity of bias gradient of last 6(4) layers exceeds 0.5 with $T = 500/50$ ($T = 20$). These results provide further evidence that EP is equivalent to BP under the condition of weak feedback.

We further studied the influence of $\beta_i$ on the cosine similarity. Figure S10(g) shows that larger $\beta_i$ leads to lower similarity of states. Figure S10(h) shows that lower $\beta_i = 0.01$ also leads to the decrease in similarity, which may caused by insufficient precision of data storage (float32 by default). Therefore, we use datatype float64 to repeat experiments. Figure S10(k,l) shows that the similarity of gradient signal remains around 1 with $\beta_i = 0.1$. This indicates that weak feedback does indeed lead to an exponential decline in gradient signals, thus requiring higher relative accuracy.

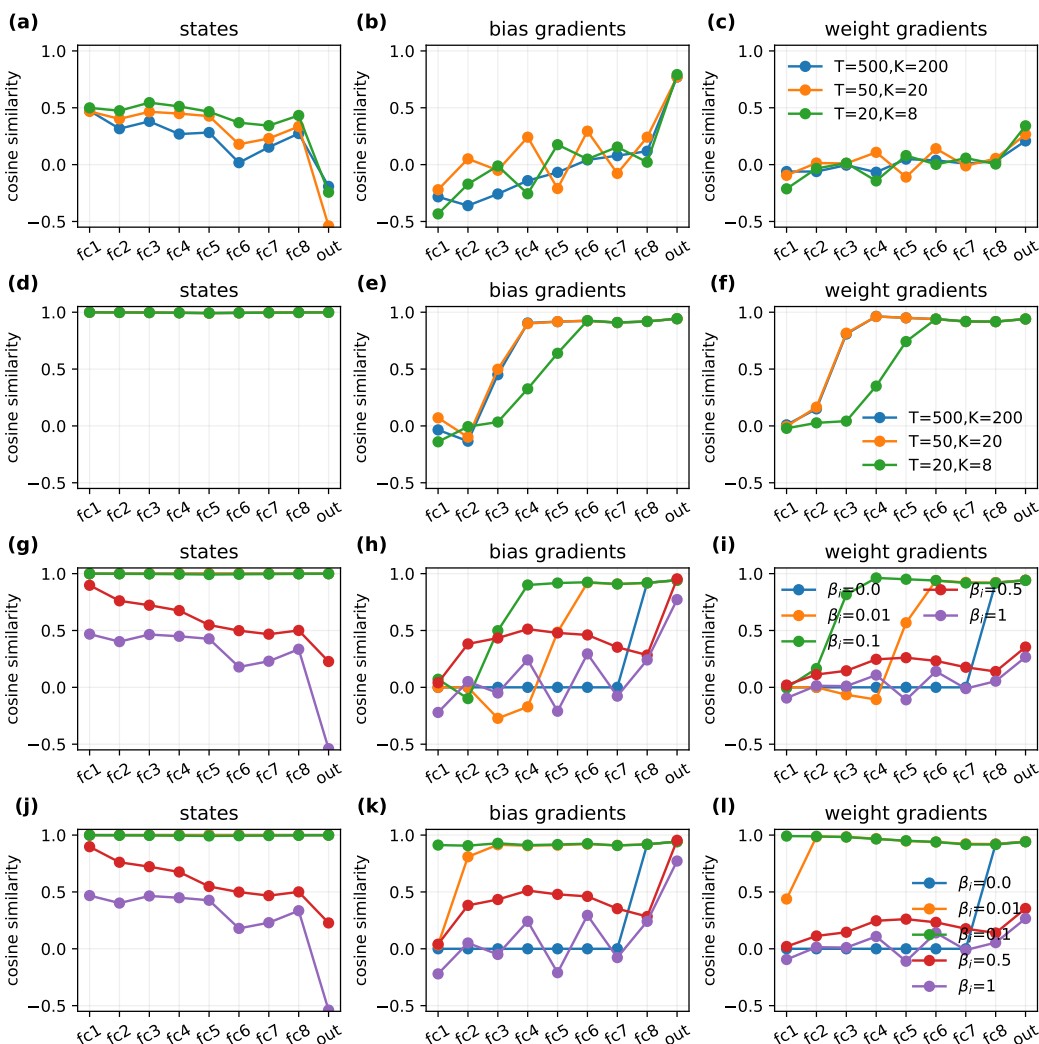

Figure S10: The cosine similarity of gradients and states between our model and feedforward model trained by BP in an 8-hidden-layer FNN (states: $s_i^0$; bias gradients: $ds_i$; weight gradients: $\Delta W_i$). The axis $x$ is the layers of the model. Error propagates from the last layer "out" to the first hidden layer "fc1" layer by layer. (a-c), with different numbers of iterations under feedback scaling $\beta_i = 1$. (d-f), with different numbers of iterations under small feedback scaling $\beta_i = 0.1$. (g-i), with different feedback scaling (T=50,K=20). (j-l), Repeat (g-i) with datatype float64 (float32 by default).

## C.5 VERIFYING THE EFFECTIVENESS OF WEAK FEEDBACK IN EQUILIBRIUM PROPAGATION

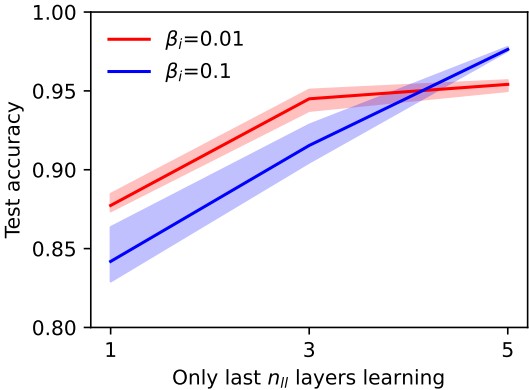

Figure S11: The testing accuracy on MNIST with different $\beta_i$ varies with $n_{ll}$. The experiments are repeated 5 times.

To demonstrate that the lower few layers of our model are indeed receiving meaningful credit signals, we report the test accuracy of only updating the last $n_{ll}$ layer (i.e., freezing the weights of Layers $1 - n_{ll}$) in Figure S11. For a 5-hidden layer model with $\beta_i = 0.1$, updating only the final layer yields a test accuracy of about 85%. As $n_{ll}$ increases to 5, the accuracy also reaches around 97.5%. A similar trend is observed for the model with $\beta_i = 0.01$. These results show that achieving over 97% accuracy requires effective gradient propagation to all layers, confirming that our model successfully delivers usable credit signals throughout the entire network.

## C.6 ROBUSTNESS TO THE NOISE

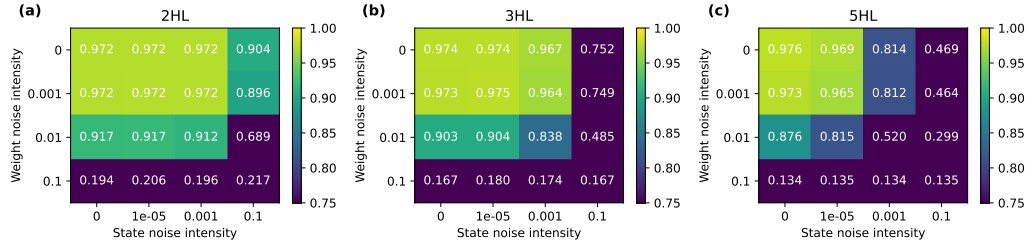

Figure S12: The maximum test accuracy model with different noise intensity on weights and states added both in training and test. (a) With 2 hidden layers; (b) With 3 hidden layers; (c) With 5 hidden layers. The model is trained for 50 epochs and the experiments are repeated 5 times.

To evaluate the robustness of the model, we introduce noise on weights and time-varying noise on states, which are random Gauss noise imposed at each weight update or at each state update, respectively. The noise on weights is directly added to the weight, while the noise on states is added as the bias $b$ in Equation 3.1. The mean absolute values of non-zero weights and neural activations after noiseless training are approximately 0.09 and 0.76 respectively.

The accuracy of the model with two hidden layers varies with two types of noises as presented in Figure S12(a). It maintains satisfactory performance when the standard deviation of state noise reaches 0.1 or the standard deviation of weight noise reaches 0.01. In deeper structures (Figure S12(b,c)), the results are consistent with the aforementioned observations for weight noise, demonstrating excellent robustness. However, the tolerance to time-varying state noise degrades significantly, which we attribute to the layer-wise noise accumulation and the distortion of weak gradient signal by the noise in the training process. To confirm our hypothesis, we impose the noises only in the test,



Figure S13: The maximum test accuracy model with different noise intensity on weights and states (the state noise is added only in test). (a) With 2 hidden layers; (b) With 3 hidden layers; (c) With 5 hidden layers. The model is trained for 50 epochs in a single experiment.

and the test accuracy almost remains unaffected (Figure S13). Therefore, the network is potentially resilient to noise. However, how to improve resilience in the training process requires further study.

## D  TRAINING DETAILS

Table S2 provides the parameters of the Adam optimizer that are used in Tables S1–S2 (Kingma & Ba, 2015). The training details for Table 1 are given in Table S3. For convolutional architectures in EP, the training process can be described by Algorithm S2. The training sample is fed into the network through $\text{Conv}_0$. Then the state of the first layer goes through max pooling $\text{MaxPool}_1$ and convolution $\text{Conv}_1$ sequentially to reach the second layer. The second layer also feedbacks its states to the first layer through transposed convolution $\text{ConvT}_1$ and max-unpooling $\text{MaxUnpool}_1$. With $T$ iterations, the RNN converges to the steady states and produces outputs through $\text{MaxPool}_2$ and a fully connected layer. Then the prediction error is computed and used to nudge the RNN by the reverse of the fully connected layer and max-unpooling $\text{MaxUnpool}_2$. Note that the unpooling $\text{MaxUnpool}_i$ requires the indices from the corresponding pooling $\text{MaxPool}_i$.

For Table S2, Adam optimizer is used for all experiments. The activation functions sigmoid-s and hard-sigmoid are defined as $\rho(x) = \frac{1}{1+e^{-4(x-0.5)}}$, $\rho(x) = \max(\min(x,0),1)$, respectively (Ernoult et al., 2019). For 5-HL, 10-HL and 20-HL architectures, the Adam optimizer parameters are as shown in Table S2 (epoch: 50, batch size: 500). The inference details of the architecture shown in Figure 3b are described by Algorithm S3. The details for convolutional architectures are given in Table S3 and Figure S14–S15. The cosine-annealing scheduler is used in convolutional architectures for CIFAR-10 ($T_{\max} = 50$, $\eta_{\min} = 10^{-6}$).

For MNIST, no pre-processing is used. For the CIFAR-10 dataset, we follow ref. (Scellier et al., 2023) to pre-process the images. We normalize the input images using mean $\mu = (0.4914, 0.4822, 0.4465)$ and standard deviation $\sigma = 3 \times (0.2023, 0.1944, 0.2010)$.

The results for comparison of time consumption were obtained in a virtualized Windows 11 environment with Intel Xeon Gold 6238R CPU, 16 GB RAM, and Nvidia RTX A5000 (24 GB VRAM). Other results were obtained on a Windows 11 environment with Intel Core i5-12490F, 32 GB RAM, and Nvidia GTX 1650 (4 GB VRAM) or a Windows 11 environment with AMD R7-7700, 32 GB RAM, and Nvidia RTX 4070 (12 GB VRAM). The default numerical precision is float32 (single-precision float).

Table S2: The parameters of the Adam optimizer.

| Parameter Name | Default Value |
|---|---|
| Learning rate (MNIST / CIFAR-10) | $10^{-3}/2 \times 10^{-4}$ |
| First-order moment estimation decay rate ($\beta_1$) | 0.9 |
| Second-order moment estimation decay rate ($\beta_2$) | 0.999 |
| Small constant for numerical stability ($\epsilon$) | $10^{-8}$ |

Table S3: Training details for Table 1 and Table 2. The results of EB-EP and P-EP come from previous work (Ernoult et al., 2019). SGD refers to Stochastic Gradient Descent with mini-batches.

| Architecture | Training approach | Optimizer | Epoch / Batch size -T/K | Learning rate | Weight decay |
|---|---|---|---|---|---|
| 2HL | P-EP (sigmoid-s) | SGD | 50/20-100/20 | $[0.005, 0.05, 0.2]$ | None |
| | Proposed (tanh, Adam) | Adam | 50/500-10/10 | $[0.001, 0.001, 0.001]$ | None |
| 3HL | P-EP (sigmoid-s) | SGD | 100/20-180/20 | $[0.002, 0.01, 0.05, 0.2]$ | None |
| | Proposed-DLR (tanh) | SGD | 100/20-18/10 | $[0.002, 0.01, 0.05, 0.2]$ | None |
| | Proposed (tanh) | SGD | 100/20-18/10 | $[0.1, 0.1, 0.1, 0.1]$ | None |
| | Proposed (tanh, Adam) | Adam | 50/500-18/10 | $10^{-3}$ | None |
| | BP (tanh, Adam) | Adam | 50/500-1/1 | $10^{-3}$ | None |
| Conv (Table 1) | P-EP (hard-sigmoid) | SGD | 40/20-200/10 | $[0.015, 0.035, 0.15]$ | None |
| | Proposed (hard-sigmoid) | SGD | 40/128-20/10 | $[0.15, 0.35, 0.9]$ | $10^{-5}$ |
| | BP (hard-sigmoid) | SGD | 40/128-1/1 | $[0.001, 0.02, 0.4]$ | $10^{-5}$ |
| Conv (Table 2 MNIST) | Proposed (hard-sigmoid) | Adam | 40/128-20/10 | $2 \times 10^{-4}$ | $10^{-6}$ |
| | BP (hard-sigmoid) | Adam | 40/128-1/1 | $2 \times 10^{-4}$ | $10^{-6}$ |
| Conv (Table 2 CIFAR-10) | Proposed (hard-sigmoid) | Adam | 50/128-40/10 | $2.5 \times 10^{-4}$ | $2 \times 10^{-4}$ |
| | BP (hard-sigmoid) | Adam | 50/128-1/1 | $2.5 \times 10^{-4}$ | $2 \times 10^{-4}$ |

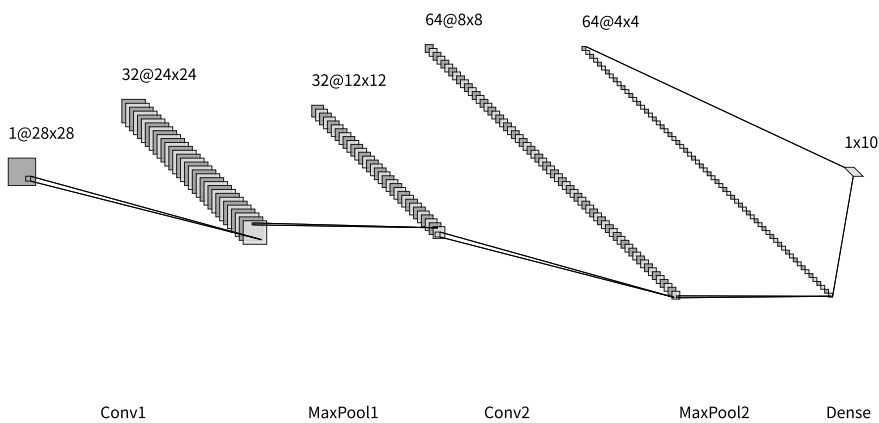

Figure S14: Convolutional architectures for MNIST.

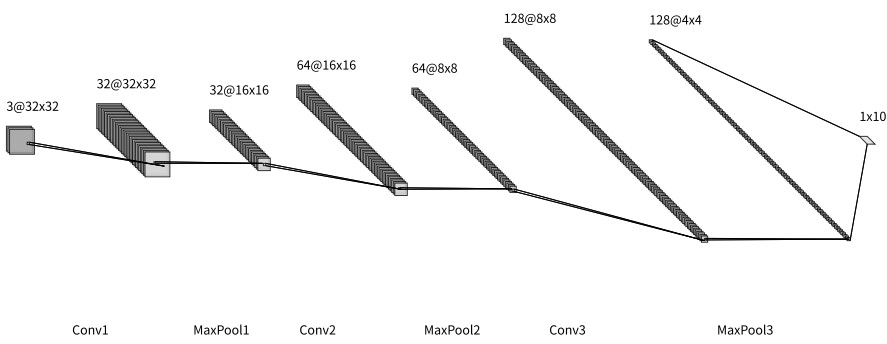

Figure S15: Convolutional architectures for CIFAR-10.

---

**Algorithm S2** Two phases in EP training process for convolution architecture

---

**Input:** Sample-label pairs $(x, s_{tar})$
**Parameter:** $\theta = [W_0, W_1, W_f, B_f, B_1, \alpha_1, \beta_1, \beta_{f1}]$
**Output:** $\theta$

 1: **function** FIRST-PHASE$(\theta, s_{tar})$
 2:  $s_0 \leftarrow x$
 3:  **for** $t \leftarrow 1$ **to** $T$ **do**
 4:   $h_1 \leftarrow \mathrm{Conv}_0(s_0) + \beta_1 \cdot \mathrm{MaxUnpool}_1(\mathrm{ConvT}_1(s_2^0))$
 5:   $h_2 \leftarrow \mathrm{Conv}_1(\mathrm{MaxPool}_1(s_1^0))$
 6:   $h_p \leftarrow W_f \cdot \mathrm{Flatten}(\mathrm{MaxPool}_2(s_2^0))$
 7:   $s_1^0, s_2^0, s_p^0 \leftarrow \rho(h_1), \rho(h_2), \mathrm{SoftMax}(h_p)$
 8:  **end for**
 9:  $\Lambda_1 \leftarrow [s_i^0],\ i = 0, 1, 2, p$
10:  **return** $\Lambda_1$
11: **end function**

12: **function** SECOND-PHASE$(\theta, \Lambda_1, s_{tar})$
13:  $s_0, s_1^{\beta_{f1}}, s_2^{\beta_{f1}}, s_p^{\beta_{f1}} \leftarrow x, s_1^0, s_2^0, s_p^0$
14:  **for** $t \leftarrow 1$ **to** $K$ **do**
15:   $e_p \leftarrow s_{tar} - s_p^{\beta_{f1}}$
16:   $h_1 \leftarrow \mathrm{Conv}_0(s_0) + \beta_1 \cdot \mathrm{MaxUnpool}_1(\mathrm{ConvT}_1(s_2^{\beta_{f1}}))$
17:   $h_2 \leftarrow \mathrm{Conv}_1(\mathrm{MaxPool}_1(s_1^{\beta_{f1}})) + \beta_f \cdot \mathrm{MaxUnpool}_2(\mathrm{Unflatten}(W_f^\top e_p))$
18:   $h_p \leftarrow W_f \cdot \mathrm{Flatten}(\mathrm{MaxPool}_2(s_2^{\beta_{f1}}))$
19:   $s_1^{\beta_{f1}}, s_2^{\beta_{f1}}, s_p^{\beta_{f1}} \leftarrow \rho(h_1), \rho(h_2), \mathrm{SoftMax}(h_p)$
20:  **end for**
21: **end function**

---

---

**Algorithm S3** EP with feedback scaling and residual connections (Figure 3b)

---

**Input:** $(x, s_{tar})$
**Parameter:** $\theta = [W_0, W_i, W_f, B_f, B_i, \beta_i, \beta_{f1}]$
**Output:** $\theta$

1: **function** ITERATION($\theta, \Lambda_1, s_{tar}$)
2:     **for** $t \leftarrow 1$ **to** $K$ **do**
3:         **if** Nudging Phase **then**
4:             $\beta_f \leftarrow \beta_{f1}$
5:             $s_0, s_1^{\beta_{f1}}, s_2^{\beta_{f1}}, s_p^{\beta_{f1}} \leftarrow x, s_1^0, s_2^0, s_p^0$
6:         **else**
7:             $\beta_f \leftarrow 0$
8:             $s_0 \leftarrow x$
9:         **end if**
10:         $h_1 \leftarrow W_0 s_0 + \beta_1 B_1 s_2^{\beta_f} + \beta_{4,1} B_{4,1} s_4^{\beta_f}$
11:         $h_2 \leftarrow W_1 s_1^{\beta_f} + \beta_2 B_2 s_3^{\beta_f}$
12:         $h_3 \leftarrow W_2 s_2^{\beta_f} + \beta_3 B_3 s_4^{\beta_f}$
13:         $h_4 \leftarrow W_3 s_3^{\beta_f} + \beta_{14} B_4 s_5^{\beta_f} + W_{1,4} s_1^{\beta_f} + \beta_{7,4} B_{7,4} s_7^{\beta_f}$
14:         $h_5 \leftarrow W_4 s_4^{\beta_f} + \beta_5 B_5 s_6^{\beta_f}$
15:         $h_6 \leftarrow W_5 s_5^{\beta_f} + \beta_6 B_6 s_7^{\beta_f}$
16:         $h_7 \leftarrow W_6 s_6^{\beta_f} + \beta_7 B_7 s_8^{\beta_f} + W_{4,7} s_4^{\beta_f} + \beta_{10,7} B_{10,7} s_{10}^{\beta_f}$
17:         $h_8 \leftarrow W_7 s_7^{\beta_f} + \beta_8 B_8 s_9^{\beta_f}$
18:         $h_9 \leftarrow W_8 s_8^{\beta_f} + \beta_9 B_9 s_{10}^{\beta_f}$
19:         $h_{10} \leftarrow W_9 s_9^{\beta_f} + \beta_f B_f e_p^{\beta_f} + W_{7,10} s_7^{\beta_f}$
20:         $h_p \leftarrow W_f s_{10}^{\beta_f}$
21:         $s_i^{\beta_f} \leftarrow \rho(h_i), \quad i = 0, 1, 2, \ldots, 10$
22:         $s_p^{\beta_f} \leftarrow \text{SoftMax}(h_p)$
23:     **end for**
24: **end function**

---

