# OpenReview forum: "Toward Practical Equilibrium Propagation: Brain-inspired Recurrent Neural Network with Feedback Regulation and Residual Connections"
_ICLR.cc/2026/Conference — ICLR 2026 Poster_

### Official Review · Reviewer_iwsx · 2025-10-16

**Soundness:** 3
**Presentation:** 3
**Contribution:** 2
**Rating:** 6
**Confidence:** 3

**Summary:**

This paper proposes a feedback-regulated residual recurrent neural network (FRE-RNN) to address the instability, slow convergence, and high costs of Equilibrium Propagation, a biologically plausible alternative to backpropagation that uses RNN dynamics for credit assignment. By scaling down feedback strength to accelerate settling and adding residual connections to counter vanishing gradients in deep networks, the approach draws from brain-like cortical modulation and recursive topologies. Evaluated on MNIST and CIFAR-10, FRE-RNN achieves BP-comparable accuracy with orders of magnitude speedups and enhanced stability.

**Strengths:**

- FRE-RNN shows practical improvements to EP while matching backpropagation accuracy on MNIST.
- The paper includes systematic ablations on hyperparameters such as T, K, β_i, α_i, and network depth. Comparisons in Table 1 and Figure 6 illustrate key trade-offs.
- This work advances energy-efficient computing suitable for neuromorphic hardwares.

**Weaknesses:**

- Individual components like feedback scaling and residual connections are not novel, and the biological framing is rather superficial.
- Evaluations are limited to MNIST and CIFAR-10 datasets. There are no tests on sequential or natural language processing (NLP) tasks. Can the authors demonstrate the proposed method on at least one sequential processing / NLP tasks?

**Questions:**

See weaknesses.

---

> ### Author Response · Authors · 2025-11-19
>
> We thank Reviewer for the critical comments. Admittedly, the performance and applicability of biologically plausible learning algorithms are far from BP. However, this line of research is not meant to replace BP, but to offer and alternative learning method for neuromorphic hardware that aims to compute with energy efficiency comparable to the brain. Moreover, it potentially provides an understanding of how biological brain works [Salvatori et al., 2022, NIPS]. This is a long shot but rewarding endeavor.
>
> ## On novelty.
> > "Individual components like feedback scaling and residual connections are not novel, and the biological framing is rather superficial."
>
> While feedback scaling and residual connections have been studied independently, their joint role within Equilibrium Propagation (EP) has not been explored. In EP, weak feedback dramatically accelerates convergence but simultaneously causes vanishing credit signals in deeper layers. The residual paths counteract this effect, enabling EP to train deeper structures that challenged previous work. The novelty of our work lies in the network structure design and the network property regulation to solve the convergence speed, a key bottlenecks of EP.
>
> Our biological framing is largely set by the background of our work. The connectivity patterns and learning mechanisms of cortical circuits have been extensively studied, because of their relevance to contemporary machine learning techniques [Yamins & DiCarlo, 2016, Nature Neuroscience; Salvatori et al., 2022, NIPS; Achterberg et al., 2023, Nature Machine Intelligence; Holk & Mejias, 2024, Current Opinion in Behavioral Sciences]. The topological characteristics of nervous system (e.g., average shortest path) are in line with the ResNet [He et al., 2016, CVPR], while weak feedback for credit assignment has been connected to regulatory mechanisms in neural circuits [Semedo et al., 2022, Nature Communications; Millidge et al., 2023, ICLR]. These are the prelude of our work. We successfully incorporated these biologically inspired features to EP. Our results not only significantly promote the applicability of EP itself, but also provide insights to the stability issue and credit assignment mechanisms in the brain. That is why we believe a biological framing is sensible.
>
> ## Sequence / NLP tasks.
> > "Evaluations are limited to MNIST and CIFAR-10 datasets. There are no tests on sequential or natural language processing (NLP) tasks. Can the authors demonstrate the proposed method on at least one sequential processing / NLP tasks?"
>
> Naturally converging RNNs trained by EP is not intended for performing complex sequence tasks [Ernoult et al., NeurIPS’19; Scellier et al., NeurIPS’23; Laborieux & Zenke, ICLR’24; Costa & Santos, Mathematics’25]. Although EP has been applied to sequence learning in specific Modern Hopfield architectures [Bal & Sengupta, IJCAI’23], such models rely on specially designed energy functions and differ fundamentally from the iterative convergence dynamics of standard RNNs. Extending naturally converging EP systems to complex sequential or NLP tasks remains an open challenge in the field rather than a limitation specific to our model. We have clarified the point in the revision.
>
> We hope the motivation behind our biological framing become clearer now and we are open to further discussion. We appreciate the critical comments that sharpen our thought.

---

> > ### Comment · Reviewer_iwsx · 2025-11-24
> >
> > Thanks for the clarification; both points make sense. I'm maintaining my score at 6.

---

### Official Review · Reviewer_7T1F · 2025-10-30

**Soundness:** 2
**Presentation:** 1
**Contribution:** 3
**Rating:** 4
**Confidence:** 4

**Summary:**

This paper proposes an improved network architecture for training with equilibrium propagation. This new architecture includes scaling down of feedback signals to improve convergence speed and, to prevent vanishing gradients caused by the weaker feedback, residual connections in both forward and backward direction.

The contributions in this paper are indeed important steps in making equilibrium propagation more practically feasible, as it's applicability is often hampered by excessively long training times and the restriction to small networks.
However, in the current state I can not recommend acceptance of this paper, for two main reasons. First, the presentation of the methods and results in the paper lacks clarity in many places (see details below in the "Questions" segment), which hampers the understanding and the applicability of the presented methods for others. Second, the experimental evidence is limited and a crucial sanity-check/ablation study is missing (see details below in the "Weaknesses" segment).

If these concerns are sufficiently addressed in the rebuttal, I am willing to increase my score of the paper.

**Strengths:**

- Making the dynamics in EP converge faster by scaling down the feedback signals is an important contribution, as the slow training times are a big drawback of the standard EP methods.
- Adding the residual connection is a good and, as far as I am aware, novel idea. It directly tackles the problem of vanishing gradients/error signals in lower layers, which is made worse by the weak feedback used to speed up convergence. Therefore, the paper immediately provides a solution to a drawback of its proposed method.

**Weaknesses:**

Experimental evidence:
- Networks with one hidden layer can solve MNIST. If a network with more hidden layers is trained on MNIST (or in fact even if the single hidden layer is wide enough), it can happen that only the weights of the top layer need to adjust to produce very high accuracies (> 95%). Therefore, just achieving high MNIST accuracies in a big (> 1 layer and > 200 neurons) does not actually prove that meaningful error signals are reaching the lower layers (which is the point of an algorithm aiming to approximate backprop).
- I suggest the following (in my opinion crucial) ablation study to remove this doubt:
	- Train a network normally (like you did) and observe the resulting distributions of the weights in each layer.
	- Initialize a new network randomly according to those distributions.
	- Freeze the weights in the lower layers and train the others.
	- Compare accuracies. If accuracies reached in this ablated model are significantly lower than in the fully trained case, we can be convinced that meaningful error signals reach the lower layers.
- The used networks have too many layers, more than are needed to solve the task. This is illustrated by the fact that also in the BP networks the most shallow ones have the best accuracy (even though they have the fewest parameters). Due to this heavy overparameterization I am not sure if any real conclusions can be drawn from these results, because the lower layers were never needed to actually solve the tasks: For example in Sec. 4.3 the paper concludes  "Here, we found that due to gradient differences across different layers induced by weak feedback, a 3-hidden-layer RNN at βi = 0.01 (Table 1, ‘ours (tanh)’) learns well with a uniform learning rate", i.e. the weight updates for the lowest layers do not need to be amplified by a higher learning rate. At the same time, the next section states that with increasing depth, gradients are weaker and weaker due to the attenuation via beta. An explanation in which the two statements do not contradict is: it did not matter if the lower layers only received a very weak error signal, because they were unnecessary in the first place.
- The paper claims in prominent places (second sentence of the abstract, last point in the contributions and final sentence of the paper) to be a strong candidate for applicability on neuromorphic hardware and to be biologically plausible. In order to be able to claim this, in my opinion, it is necessary to prove that this algorithm deals well with noise (which is prevalent in both biological and neuromorphic systems). In particular it needs to be shown, that weight quantization and/or noise on weights as well as time-varying noise on the state variables s do not substantially impact functionality.

Literature:

- Weak feedback has been a prominent feature in other bio-plausible backpropagation algorithms, this should be acknowledged [1, 2, 3]

[1] Sacramento, João, et al. "Dendritic cortical microcircuits approximate the backpropagation algorithm." _Advances in neural information processing systems_ 31 (2018).

[2] Haider, Paul, et al. "Latent equilibrium: A unified learning theory for arbitrarily fast computation with arbitrarily slow neurons." _Advances in neural information processing systems_ 34 (2021): 17839-17851.

[3] Meulemans, Alexander, et al. "Credit assignment in neural networks through deep feedback control." _Advances in Neural Information Processing Systems_ 34 (2021): 4674-4687.

**Questions:**

Areas of the paper lacking clarity:

- Fig 1, Eq 2 and surrounding text contradicting each other:
	- The text around Eq 2, the surrounding inline equations and the referenced Fig 1 describe a 2-hidden-layer setting (explicitly mentioned in the text) but Eq 2 does not match that, it seems to describe a single layer setting. This is confusing.
	- The quantity alpha which is part of the model depicted in Fig 1 is missing in the Eq 2, it is not clear for which of the forward weights this scaling is really applied and for which it is not?
- Recurrency in the network (Eqn 1, 2 vs Alg. 1):
	- Eqs 1 + 2 feature an explicit recurrency within the layer. In Eq 2 this is expressed via the parameter W. However, Alg. 1 and Fig 1 do not feature those weights.
	- What about the learning of the layer-internal weights W? Are they not adjusted? How are they chosen if they are fixed?
	- Do your networks contain layer-internal recurrent weights W or not?
- Fig 1b: the depicted network structure is not helpful to actually understand the CNN-based setup. Could there be a more extensive illustration, e.g. in the appendix?
- Fig 2:
	- What is the value denoted by color? The state variable s? If yes of which layer? A label on the color bar would help.
	- If two color bars for left and right columns would be used, the actual values would be better visible (the two columns have different ranges of values, so if each gets a colorbar more details are visible).
	- Caption: "The hidden layer neurons are numbered from input to output." What does that mean? How can they be arranged, it's a recurrently connected set of neurons? What makes one closer to the input and another closer to the output?
- Choices/description of experimental setup unclear:
	- l. 198-200: How exactly is the comparison with BP and FA done? Is the same network architecture trained with BP or FA (i.e. BP through time) or is a vanilla ANN of the same feed-forward sizes used?
	- Section 3.3: Why can't you have AGT for symmetric setups or regularly spaced residuals like in Fig3a for asymmetric setups?
	- Why do the numbers in Fig 5 not match the numbers in Tab 1?
	- Did the setups in Fig 4 + 5 have skip connections?
	- Tab1: Why are there no BP results for the 2HL case?
	- Tab2: According to the statement that T = 10 x N_hidden was used, the convergence times set for these results is much higher than anywhere else in the paper. Why? Also, this should be at least mentioned as a qualifier in the text.
- Fig 6:
	- What is the network size (layer widths not given)?
	- What is K?
	- Caption: "By default T = 10 * N_hidden" Does that really mean that you are comparing networks of different sizes here? And that in a - d) the red network has a hidden layer that consists of 1 neuron?
- Comparison of accuracies in Figures 4 and 6:
	- When accuracies are plotted for multiple different models it is very difficult to actually see the final accuracies (and their differences) in the plots. It would be much easier to see if errors instead of accuracies were plotted and the y-axis would be on a log-scale.
	- Fig 6: Additionally, it would be much easier to interpret, if the plots shared the same y-axis range.

---

> ### Author Response · Authors · 2025-11-19
>
> # Reply 1/2
> We thank the reviewer 7T1F30 for taking time to thoroughly review our work. The details comments have greatly helped improve our manuscripts. Below are our replies to the comments on the weaknesses and answers to the questions
>
> ## Experimental evidence:
> > Ablation study to prove that meaningful error signals are reaching the lower layers
>
> We have conducted the suggested ablation study exactly as suggested. The accuracy of the “frozen lower layers” model dropped by 3%-13% compared to our full model (Figure S11 in Appendix C.5). It shows that the network performance is significantly better when lower layers are trainable, suggesting meaningful error signals have been propagated to the lower layers.
>
> > "The used networks have too many layers, more than are needed to solve the task."
>
> We fully agree with the reviewer that to solve the MNIST do not need deep networks. The purpose of using deeper models is to demonstrate that our structural modifications facilitate training deeper networks [Nøkland, 2016, NIPS].
>
> > "Noise on weights as well as time-varying noise on the state variables"
>
> We have performed the suggested experiments with weight noise and time-varying state noise (Appendix C.6). At each weight/state update, a certain level of Gaussian noise is superimposed on the weights or states. The shallow models exhibit excellent noise robustness, while the tolerance for state noise decreases for the deeper models (Figure S12). We attribute the susceptibility to noise of deeper model to distorted gradient information. However, when noise is only applied in test process, the accuracy remains almost unchanged (Figure S13), suggesting the network’s potential noise resilience. However, improving noise resilience during training requires further research.
>
> ## Literature.
> We have added related papers about weak feedback to our paper.
>
> ## Questions.
> > "Fig 1, Eq 2 and surrounding text contradicting each other. \
> The text around Eq 2, the surrounding inline equations and the referenced Fig 1 describe a 2-hidden-layer setting (explicitly mentioned in the text) but Eq 2 does not match that, it seems to describe a single layer setting. This is confusing.\
> The quantity alpha which is part of the model depicted in Fig 1 is missing in the Eq 2, it is not clear for which of the forward weights this scaling is really applied and for which it is not?
> "
>
> Eq. 2 describe the dynamics of the whole RNN, not specific to one of the layers, i.e., $s^{\beta_f}=[s^{\beta_f}_1, s^{\beta_f}_2]$, weight $W$ includes implicitly scaling $\alpha_i$, $\beta_i$ and weights $W_i$ and $B_i$. $W_0 s_0$ is the input of RNN and whole model, not the internal dynamics of RNN, thus without scaling $\alpha_i$. We have claritied this point in the revision.
>
> Algorithm 1 describes the practical implementation and provide details on how the scaling is applied.
>
> > "Recurrency in the network (Eqn 1, 2 vs Alg. 1)
> Eqs 1 + 2 feature an explicit recurrency within the layer. In Eq 2 this is expressed via the parameter W. However, Alg. 1 and Fig 1 do not feature those weights.\
> What about the learning of the layer-internal weights W? Are they not adjusted? How are they chosen if they are fixed?\
> Do your networks contain layer-internal recurrent weights W or not?
> "
>
> There are no connections within a single hidden layer. The whole RNN internal weight $W$ is composed of weights $W_1$, $B_1$ and scaling $\alpha_1$, $\beta_1$, i.e., $W = [ [0, \alpha_1 W_1]; [\beta_1 B_1, 0] ]$.
>
> > "Fig 1b: the depicted network structure is not helpful to actually understand the CNN-based setup. Could there be a more extensive illustration, e.g. in the appendix?"
>
> We provide the algorithm and architectures of the CNN implementation in Appendix D, which is similar to the setting of ref. [Laborieux and Zenke, 2024, ICLR; Ernoult et al., 2019, NIPS].

---

> ### Author Response · Authors · 2025-11-19
>
> # Reply 2/2
>
> > "Fig 2:\
> What is the value denoted by color? The state variable s? If yes of which layer? A label on the color bar would help.\
> If two color bars for left and right columns would be used, the actual values would be better visible (the two columns have different ranges of values, so if each gets a colorbar more details are visible).\
> Caption: "The hidden layer neurons are numbered from input to output." What does that mean? How can they be arranged, it's a recurrently connected set of neurons? What makes one closer to the input and another closer to the output?
> "
>
> Colors indicate neuronal activity (Figure 2a,c,e,g) and changes in activity (Figure 2b,d,f,h). We index all neurons in the two hidden layers shown in Fig. 1 with numbers, which serve as the y-axis in Fig. 2. Neurons in the first hidden layer are indexed 0–63, and those in the second hidden layer 64–127.
>
> As suggested, we have adopted two distinct color bars to enhance the visualization of the state and state differences.
>
> There are no connections within a single hidden layer. Getting closer to the input and closer to the output refers to the position of the hidden layer. We have revised the text to avoid ambiguity.
>
>
> > "Choices/description of experimental setup unclear:\
> l. 198-200: How exactly is the comparison with BP and FA done? Is the same network architecture trained with BP or FA (i.e. BP through time) or is a vanilla ANN of the same feed-forward sizes used?\
> Section 3.3: Why can't you have AGT for symmetric setups or regularly spaced residuals like in Fig3a for asymmetric setups?\
> Why do the numbers in Fig 5 not match the numbers in Tab 1?\
> Did the setups in Fig 4 + 5 have skip connections?\
> Tab1: Why are there no BP results for the 2HL case?\
> Tab2: According to the statement that T = 10 x N_hidden was used, the convergence times set for these results is much higher than anywhere else in the paper. Why? Also, this should be at least mentioned as a qualifier in the text.
> "
>
> A vanilla ANN of the same feedforward size trained by backpropagation is used for comparison.
>
> Because in the AGT setting, feedback connections are randomly generated—meaning feedforward and feedback connections are necessarily asymmetric. So, AGT is by our definition not symmetric.
>
> For the convenience of horizontal comparison with other papers, the experiments in Table 1 adopt the setting of 512 neurons per hidden layer—thus its accuracy significantly exceeding the default configuration (64 neurons per hidden layer) used elsewhere in this paper.
>
> No, residual connections are not adopted unless otherwise specified in Table 2.
>
> We have now added BP results of 2HL to Table 1 and the conclusions remain the same.
>
> We have revised $N_{hidden}$ to $N_{Hidden Layer}$ to avoid ambiguity. As the number of hidden layers increases, the network becomes difficult to converge and requires more iterations. $T=10\times N_{Hidden Layer}$ is an empirical number that ensures convergence.
>
>
> > "Fig 6:
> What is the network size (layer widths not given)?\
> What is K?\
> Caption: "By default T = 10 * N_hidden" Does that really mean that you are comparing networks of different sizes here? And that in a - d) the red network has a hidden layer that consists of 1 neuron?
> "
>
> We use the default configuration (64 neurons per hidden layer). We have added this information to the caption.
>
> $K$ indexes iteration steps of the second phase. It has been defined in Section 3.1.
>
> $N_{hidden}$ has been revised to $N_{Hidden Layer}$ now.
>
>
> > "Comparison of accuracies in Figures 4 and 6:\
> When accuracies are plotted for multiple different models it is very difficult to actually see the final accuracies (and their differences) in the plots. It would be much easier to see if errors instead of accuracies were plotted and the y-axis would be on a log-scale.\
> Fig 6: Additionally, it would be much easier to interpret, if the plots shared the same y-axis range.
> "
>
> Following the reviewers’ suggestions, we have revised Figs. 4 and 6. The revised figures are clearer. we appreciate the valuable suggestions.
>
> We thank the reviewer again for the helpful comments. Beside the specified modifications, we have reworded our text at other places relevant to the raised questions. We believe the revisions have addressed all raised concerns.

---

> > ### Comment · Reviewer_7T1F · 2025-11-28
> > **Reply to rebuttal**
> >
> > I thank the authors for their thorough care when addressing my comments.
> > The new results and the improved clarity strengthen the paper significantly.
> >
> > I have only one remaining question (as it is important to actually judge the noise robustness):
> > You quantify the amount of noise by the standard deviation of the Gaussian noise that is applied but it is also important to know what the value range of the quantity the noise is applied to is. For example, applying noise with a std-dev of 0.01 has a completely different effect on a quantity that is in a value range of -1000 to a 1000 and another that is in the range of -0.1 and 0.1. I think it is therefore very important to accompany the noise robustness evaluations and their interpretation with estimates of the value ranges of the noised quantities such that the relative strength of the noise can be judged.

---

> > > ### Author Response · Authors · 2025-11-29
> > >
> > > We appreciate the reviewer's thoughtful question. We have included the mean absolute values of weights and neural states in Appendix C.6, which are 0.09 and 0.76, respectively. These values are computed after training in the absence of noise.  We thank the reviewer again for the time and care dedicated to reviewing our manuscript.

---

### Official Review · Reviewer_TKZt · 2025-10-31

**Soundness:** 2
**Presentation:** 1
**Contribution:** 2
**Rating:** 2
**Confidence:** 4

**Summary:**

Equilibrium propagation is a biologically plausible method for credit assignment in deep neural networks. It operates through two phases, each requiring convergence to a fixed point. However, this relaxation process becomes exponentially slower as network depth increases. This paper addresses this problem by weakening feedback connections and introducing skip connections. The authors then demonstrate empirically that these modifications reduce convergence time and thus improve performance.

**Strengths:**

**Innovative approach to the relaxation time problem.** The authors address the increased relaxation time in deeper networks indirectly by effectively reducing the depth over which error signals propagate. After several multiplications by the $\beta_i$ coefficients that scale the feedback weights, negligible signal remains. The residual connections provide an intuitive mechanism for ensuring all layers receive teaching signals.

**Weaknesses:**

My main concerns are that the results are difficult to trust and the paper lacks proper analysis of what the method is actually doing.

**Insufficient analysis of the method's mechanisms.** The proposed architectural modifications can be understood as a way to learn only layers that have direct impact on the output (similar to reservoir computing). This interpretation raises several critical questions: How far in the hierarchy are accurate error signals propagated? How would the proposed method behave if only the last layer or last few layers were trained? Answering these questions is essential for scientific understanding of the proposed modification and is currently absent. As a side note, the proposed modifications are rather independent to EP, analyzing their effects on other biologically plausible learning algorithms would strengthen the analysis.

**Results are difficult to trust.** This stems from both presentation issues and technical inaccuracies:

- The dynamics of Equation (2) are not energy-based dynamics and, more importantly, do not converge to stationary points of the energy function. Consequently, the EP update will only be an approximation. This limitation is not mentioned anywhere in the paper.
- The EP framework (which receives minimal introduction) provides simple and elegant ways to describe the corresponding learning algorithm, yet the authors complicate it unnecessarily. For instance, they fix the error to its forward pass value; while correct, this adds unnecessary complexity. Algorithm 1 occupies more than half a page describing something explained concisely in the main text. Using the same symbol ($\beta$) for both feedback strength and nudging strength creates confusion. While these could be dismissed as cosmetic details, collectively they create an impression of confusion that undermines the paper.
- The typical baseline for EP is recurrent backpropagation, not standard backpropagation, since the network is recurrent. Backpropagation can only be applied to a feedforward version of the model, which is generally non-trivial to define. Consequently, the empirical comparisons feel like comparing apples to oranges.

**Additional remarks:**

- Figure 2 requires more statistically robust analysis.
- It is unclear why the authors sweep over forward connection strengths in Section 4.1, as this appears unrelated to their method.
- Figure 6 does not convincingly demonstrate that "Larger β_i requires more iterations for the RNN to reach fixed point."
- Empirical results are overly descriptive and sometimes lack proper interpretation. For example, the takeaway from Section 4.3 is unclear.

In my opinion, the paper is not currently ready for acceptance. It requires another extensive round of improvements to both the results themselves and their presentation.

**Questions:**

--

---

> ### Author Response · Authors · 2025-11-19
>
> # Reply 1/2
>
> We fully understand the reviewer’s concern. We have addressed these concerns with suggested experiments. The results provide clear and concrete evidences that corroborate our conclusions. We have revised the manuscript as detailed below.
>
> ## Analysis of the method’s mechanisms.
> > “Insufficient analysis of the method's mechanisms. The proposed architectural modifications can be understood as a way to learn only layers that have direct impact on the output (similar to reservoir computing). This interpretation raises several critical questions: How far in the hierarchy are accurate error signals propagated? How would the proposed method behave if only the last layer or last few layers were trained?”
>
> We agree that understanding how far accurate error signals propagate is essential. A RC-like learning mechanism is a sensible alternative explanation. However, it is unlikely to be our case as each layer of the network has only 64 neurons. RC usually requires large number of neurons to get high accuracy. To further rule out this possible explanation, we conducted new control experiments. The results show that training only the last layer or last few layers does not achieve competitive accuracy (Appendix C.5). Therefore, training the deeper layers is essential.
>
> > “the proposed modifications are rather independent to EP, analyzing their effects on other biologically plausible learning algorithms would strengthen the analysis.”
>
> Weak feedback and residual connections are existing approaches in machine learning community, and they have biological ground [Sacramento et al., 2018, NIPS; Haider et al., 2021, NIPS; Meulemans et al., 2021, NIPS; Gammell et al., 2021, Frontiers in Computational Neuroscience]. However, we introduce them to EP for the first time. Weak top-down signals not only improve the convergence of RNN dynamics, but also make the gradient signal more aligned to the feedforward network trained by BP (Appendix C.4). We have added more discussion in Section 5.
>
> ## Presentation and technical issues
>
> ### Clarification regarding Equation (2) and EP approximation.
> > "The dynamics of Equation (2) are not energy-based dynamics and, more importantly, do not converge to stationary points of the energy function. Consequently, the EP update will only be an approximation. This limitation is not mentioned anywhere in the paper."
>
> Energy function can only be defined for RNN with symmetric recurrent weights. The original EP is extended to vector field dynamics for RNN with asymmetric weights, as described by Equation (2) [Scellier et al., 2018, arXiv:1808.04873]. Such non-energy-based formulation has also been adopted by other following works [Ernoult et al., 2019, NIPS; Laborieux et al., 2021, Frontiers in Neuroscience; Laborieux and Zenke, 2024, ICLR], etc. We have now stressed this point in Section 3.2.
>
> ### Clarifications regarding the algorithmic presentation.
> > "they fix the error to its forward pass value; while correct, this adds unnecessary complexity.."
>
> Fixing the error to its forward pass value in Algorithm 1 line 15 is mathematically equivalent the standard EP formulation and is used explicitly in previous work [Laborieux and Zenke, 2024, ICLR] (Page 19). We followed such formulation for the convenience of comparing the gradients of our model and that of FNNs trained by BP (Appendix C.4).
>
> > "Algorithm 1 occupies more than half a page describing something explained concisely in the main text. Using the same symbol () for both feedback strength and nudging strength creates confusion. While these could be dismissed as cosmetic details, collectively they create an impression of confusion that undermines the paper."
>
> Algorithm 1 is a pseudocode detailing the algorithmic implementation, which facilitate repeating our results by other groups. However, if the reviewer believe it is unnecessary, we are also open to other suggestions.
>
> > "Using the same symbol () for both feedback strength and nudging strength creates confusion."
>
> $\beta_i$ and $\beta_f$ are similar in the sense that they are both associated with the scaling of gradient signals (implicitly or explicitly). We have stressed this similarity in our paper to clear the confusion in Section 3.2.

---

> ### Author Response · Authors · 2025-11-19
>
> # Reply 2/2
> ### Choice of baseline.
> > "The typical baseline for EP is recurrent backpropagation, not standard backpropagation, since the network is recurrent. Backpropagation can only be applied to a feedforward version of the model, which is generally non-trivial to define. "
>
> Indeed, recurrent backpropagation has been the canonical baseline for Prototypical setting of Equilibrium Propagation. It has been proved that EP can achieve performance comparable to BPTT [Ernoult et al., 2019, NIPS; Laborieux et al., 2021, Frontiers in Neuroscience]. Since we are motivated to improve EP, we think BP is a more natural baseline.
>
> We agree that defining a FNN equivalent to our model is non-trivial. With weak feedback, our model behaves more like an FNN, and its gradients approximate those of BP-trained FNNs (Appendix C.4). Since static image tasks are conventionally solved by BP-trained feedforward networks, comparing with BP-trained FNNs provides a meaningful benchmark for us.
>
> ## Additional remarks.
> > "Figure 2 requires more statistically robust analysis"
>
> Figure 2 displays the typical RNN dynamics. The statistically robust analysis is provided in Figure 5(d), which quantifies the number of iterations required for convergence under different conditions. We have clarified this point in the text in the caption of Figure 2.
>
>
> > "It is unclear why the authors sweep over forward connection strengths in Section 4.1, as this appears unrelated to their method."
>
> We understand the reviewer’s point. However, it is still relevant for understanding how does the scaling affect the network’s performance in deeper networks (please see also our reply to question 2 of reviewer Vmi301).
>
> > "Figure 6 does not convincingly demonstrate that "Larger $β_i$ requires more iterations for the RNN to reach fixed point.""
>
> We have refined the explanation. Larger $\beta_i$ indeed requires more iterations to reach a given performance level (Figs. 6b–d). We hypothesize that this arises because larger $\beta_i$ leads to larger spectral radius and larger finite time maximum Lyapunov exponent, which slows down convergence to a fixed point (Figs. 2 and 5b-d), causing inaccurate error estimation when iterations are insufficient. We now present this interpretation more clearly in Section 4.2.
>
> > "Empirical results are overly descriptive and sometimes lack proper interpretation. For example, the takeaway from Section 4.3 is unclear."
>
> We have added interpretations of the results, i.e., it suggests that the different plasticity can be realized by feedback regulation. We have also refined our take-away messages elsewhere.
>
> We are convinced that the new experiments and clarifications in the revisions have substantially enhanced our manuscript. We appreciate the reviewer’s critical and helpful comments and we are open to more discussions.

---

### Official Review · Reviewer_Vmi3 · 2025-10-31

**Soundness:** 2
**Presentation:** 3
**Contribution:** 3
**Rating:** 6
**Confidence:** 4

**Summary:**

This paper introduces FRE-RNN, a biologically inspired recurrent neural network designed to make Equilibrium Propagation (EP) practical for modern deep learning. EP is a local, brain-inspired learning rule but has traditionally suffered from slow convergence and instability in deeper architectures. To address this, the authors introduce feedback regulation, scaling the feedback weights to less than 1 to stabilize dynamics and accelerate convergence. They further incorporate residual and random skip connections to improve gradient flow and enable deeper recurrent structures. Experiments on MNIST demonstrate that FRE-RNN achieves significantly faster runtimes than standard EP while maintaining accuracy comparable to backpropagation, and that it can successfully train deeper RNNs than previous equilibrium-based methods.

**Strengths:**

This paper presents an interesting and well-motivated step toward developing biologically inspired yet computationally practical learning systems. The proposed FRE-RNN introduces effective modifications, such as feedback regulation and residual connections, that significantly improve the efficiency and stability of Equilibrium Propagation models. The authors demonstrate that their approach achieves at least an order-of-magnitude speedup in convergence while maintaining accuracy comparable to backpropagation-based networks. The experiments are clear, well-structured, and provide convincing evidence that these improvements make equilibrium-based RNNs more scalable to deeper architectures. Overall, the paper is clearly written, conceptually sound, and provides strong justification for the proposed modifications.

**Weaknesses:**

While the proposed framework demonstrates clear improvements over prior equilibrium propagation methods, it seems to only work well on relatively simple datasets. The approach performs well on MNIST but struggles on CIFAR-10, suggesting that it may not yet generalize effectively to more complex, real-world scenarios. Although the paper emphasizes the potential of bridging neuroscience and machine learning, it would benefit from a deeper discussion on the practical implications and real-world applicability of such biologically inspired models, especially given their current performance limitations. In addition, the experiments focus on image data, which is may not be the best fit for RNN-based architectures. Demonstrating results on temporal or sequential modalities, such as text, would have strengthened the case for the proposed approach. Finally, while the paper is generally well written, certain sections, particularly Section 4.1, could be improved for clarity and flow, as some results are presented in a non-linear order.

**Questions:**

1.	If the proposed model, which aims to closely mimic biological mechanisms, performs poorly on more complex datasets, what practical advantage does it offer over less biologically inspired models that achieve much higher accuracy?
2.	For the results shown in Figure 5, were the reported trends averaged over multiple runs, or are they based on a single experiment? Can the authors conclusively state that lower beta values and higher alpha values consistently lead to better performance across architectures and datasets?

---

> ### Author Response · Authors · 2025-11-19
>
> ### On weaknesses:
>
> >Performance on more complex dataset, discussion on the practical implications and real-world applicability
>
> In this work, we mainly address the training cost issue. Indeed, as displayed in Table 2, a performance gap exists between our model and BP primarily in deep architectures (e.g., 10/20 hidden layers). For networks with fewer than 5 layers—under both convolutional and layered configurations—our model performs comparably to BP on both MNIST and CIFAR-10. By combining with other structures like CNN, EP has potential to solve more complex problems [Elayedam & Srinivasan, 2025, arXiv.2509.26003].
>
> The residual connections we introduce substantially mitigate the problem of training in deeper model, with further potential for refinements. More importantly, the training cost is only one-tenth of prior EP.
> EP is not meant to replace BP; instead, it aims for learning neuromorphic and physical neural networks [Momeni et al., 2025, Training of physical neural networks, Nature 645: 53-61; Laydevant et al., 2024, Training an Ising machine with equilibrium propagation, Nat Commun 15(1): 3671.]. We believe the advancement presented in our manuscript greatly enhance the biological and physical plausibility of EP.
>
> >Temporal or sequential modalities
>
> Existing EP focuses primarily on static-input settings [Ernoult et al., NeurIPS’19; Scellier et al., NeurIPS’23; Laborieux & Zenke, ICLR’24]. Although EP has been applied to sequence learning in specific Modern Hopfield architectures [Bal & Sengupta, IJCAI’23], such models rely on specially designed energy functions and differ fundamentally from the convergence dynamics of standard RNNs. Extending our method to fully general sequence tasks is nontrivial and remains an important direction for future work. We have added a clearer discussion in the revision.
>
> >Clarity and flow of Certain sections, particularly Section 4.1
>
> We have revised Section 4.1 to improve clarity and flow. We have exchanged the positions of Figure 4 and Figure 5. Figure 5 shows the influence of scaling, which further explains the performance results in Figure 4.
>
> ### Question 1:
> > "If the proposed model, which aims to closely mimic biological mechanisms, performs poorly on more complex datasets, what practical advantage does it offer over less biologically inspired models that achieve much higher accuracy?"
>
> We see two main advantages, even though a performance gap remains in deep structures.
>
> 1.	Hardware friendliness and biological plausibility. EP is suitable for physical hardware implementations, since it avoids explicit computation of derivatives of nonlinear activation functions—a known bottleneck for physical systems. Recent works argue that local, convergence-based learning rules are promising for physical neural networks [Momeni et al., 2025, Training of physical neural networks, Nature 645: 53-61]. Our framework inherits this key advantage: it relies only on naturally emerging dynamics and local updates, making it more scalable to neuromorphic or analog substrates than BP. Besides, its performance and practical applicability remain amenable to further enhancement.
>
> 2.	Relevance to neuroscience and brain-like architectures. Training recurrent systems with structured feedback establishes an approach aligned with cortical computation. Our work can serve as a framework for understanding brain circuits or cognitive functions, in line with prior work linking goal-driven deep models with neuroscience [Yamins & DiCarlo, 2016, Nature Neuroscience 19: 356-365; Salvatori et al., 2022, NIPS; Achterberg et al., 2023, Nature Machine Intelligence 5; Holk & Mejias, 2024, Current Opinion in Behavioral Sciences 56: 101351]. Thus, biologically inspired learning mechanisms—though not matching BP on some benchmarks—offer conceptual and practical benefits beyond artificial intelligence.
>
>
> ### Question 2:
> > "For the results shown in Figure 5, were the reported trends averaged over multiple runs, or are they based on a single experiment? Can the authors conclusively state that lower beta values and higher alpha values consistently lead to better performance across architectures and datasets?"
>
> All results in Figure 5 (Figure 4 in revision) were averaged over 5 runs as stated in the caption. The range of α in the original submission was limited, which may have caused confusion. We have updated the ranges and repeated the experiments. The revised results show that there exists an optimal interval for both β and α rather than a monotonic trend.
>
> We appreciate the reviewer’s insightful comments and questions, which have greatly helped enhance our paper.

---

### Author Response · Authors · 2025-11-19
**Global response**

We sincerely thank all reviewers for their rigorous examination of our work and their valuable feedback. We have substantially revised the manuscript to address all the points raised. The reviewers' insights are instrumental in guiding the improvements of our paper.

Specifically, in response to the question on the learning mechanism, we have performed new experiments as suggested by Reviewer TKZt31 and Reviewer 7T1F30. The results clearly confirm that the network’s performance relies on effective gradient propagation to lower layers. Furthermore, we now demonstrate our model’s robustness to the noise on weights and time-varying noise on states, a point raised by Reviewer 7T1F30.

We have refined the descriptions in Section 3 and Section 4 to improve clarity and avoid potential misinterpretations noted by the reviewers. We also added a discussion on the implications of weak feedback and the applicability to sequence tasks, as pointed out by Reviewer Vmi301 and Reviewer iwsx16.

We are confident that these revisions address the reviewers’ concerns and greatly improve the quality of the manuscript. We thank the reviewers again for their time and insights, and look forward to their further assessment.

---

### Meta-Review · Area_Chair_cLN2 · 2026-01-06

**Summary:**

Two of the reviewers gave positive scores (6, 6) while two reviewers gave negative scores (2, 4).

The primary concerns were that the conclusions require further empirical evidence (several specific experiments were proposed) and presentation/technical issues make the results difficult to trust.

**Reviewer Concerns:**

The authors performed multiple experiments suggested by the reviewers to provide further empirical evidence in support of their claims. I think the concerns about the insufficiency of the empirical evidence were addressed. The authors also adequately addressed concerns about presentation/technical issues.

**Reviewer Scores:**

I believe that the reviewers who gave positive scores (6, 6) would have maintained their positive scores. I think that Reviewer 7T1F would have raised their score from a 4 to a 6. I think that Reviewer TKZt would have raised their score from a 2 to a 4.

---

### Decision · Program_Chairs · 2026-01-26

Accept (Poster)